# Multilingual Simplification of Medical Texts

**Sebastian Joseph**[1]    **Kathryn Kazanas**[1]    **Keziah Reina**[1]    **Vishnesh J. Ramanathan**[2]
**Wei Xu**[2]    **Byron C. Wallace**[3]    **Junyi Jessy Li**[1]

[1]The University of Texas at Austin

[2]Georgia Institute of Technology, [3]Northeastern University

{sebaj, kreina, jessy}@utexas.edu, katkazanas@gmail.com

{vishnesh@, wei.xu@cc.}gatech.edu, b.wallace@northeastern.edu

## Abstract

Automated text simplification aims to produce simple versions of complex texts. This task is especially useful in the medical domain, where the latest medical findings are typically communicated via complex, technical articles. This creates barriers for laypeople seeking access to up-to-date medical findings, consequently impeding progress on health literacy. Most existing work on medical text simplification has focused on monolingual settings, with the result that such evidence would be available only in just one language (most often, English). This work addresses this limitation via *multilingual* simplification, i.e., directly simplifying complex texts into simplified texts in multiple languages. We introduce MULTICOCHRANE, the first sentence-aligned multilingual text simplification dataset for the medical domain in four languages: English, Spanish, French, and Farsi. We evaluate fine-tuned and zero-shot models across these languages with extensive human assessments and analyses. Although models can generate viable simplified texts, we identify several outstanding challenges that this dataset might be used to address.

## 1 Introduction

Important findings in medicine are typically presented in technical, jargon-laden language in journal articles or reviews, which is difficult for laypeople to understand. This impedes transparent and fair access to critical medical information and ultimately hinders health literacy, which is "one of the most promising and cost-effective approaches to overcome the Non-Communicable Disease challenge" (Liu et al., 2020).

Text simplification models which automatically transform complex texts into simpler versions understandable by lay readers (Siddharthan, 2014; Alva-Manchego et al., 2020) have emerged as a promising means of providing wider access to published medical evidence. Recent work on simplification has fine-tuned large pre-trained models (Van

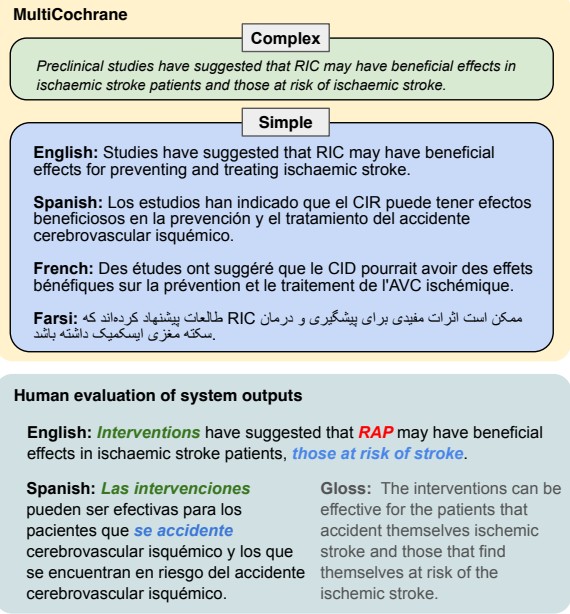

Figure 1: Examples of our dataset and system outputs for multilingual medical text simplification. The dataset has simplifications available in four languages (English, Spanish, French, Farsi), and system outputs were analyzed for factual errors (red), fluency errors (blue), and simplicity (green) among other criteria.

et al., 2020; Cardon and Grabar, 2020; Devaraj et al., 2021; Guo et al., 2022; Trienes et al., 2022; Basu et al., 2023), explored reinforcement learning (Phatak et al., 2022), and evaluated zero-shot performance via prompting (August et al., 2022).

However, this work has so far exclusively considered monolingual text simplification. Consequently, *we do not know how well models perform when the simplified text is **not** in the same language as the original, complex text*. This limits the (potential) availability of simplified information to a few high-resource languages—especially for the medical domain—and leads to equity issues such that individuals who do not speak English will still face a high barrier to information access.[1] Work

---

[1]While machine translation, to a certain extent, may be able

in this direction is impeded by a lack of data. The largest resources in text simplification are not in the medical domain, and parallel corpora for medical simplification only exist in single languages, namely English (Devaraj et al., 2021; Guo et al., 2022; Basu et al., 2023), French (Grabar and Cardon, 2018; Cardon and Grabar, 2020), and German (Trienes et al., 2022).

This paper advances the state of *multilingual medical simplification*, in which complex texts are to be directly simplified into target languages. Our contributions are as follows. We introduce MULTICOCHRANE (Figure 1), the first parallel dataset for medical text simplification across multiple languages: English, Spanish, French, and Farsi. MULTICOCHRANE contains aligned sentence pairs sourced from the Cochrane Library of Systematic Reviews,[2] which is a library of meta-analyses of treatment effectiveness.

These review articles include both a technical abstract and a plain-language summary (PLS), from which we derive the two subsets of MULTI-COCHRANE: (1) MC-CLEAN, 101 technical abstracts with expert-annotated manual alignments first derived in English, then semi-automatically aligned to other languages (partially verified by bilingual speakers). MC-CLEAN contains ∼5k sentence pairs across all 4 languages. (2) MC-NOISY, a larger but noisier subset created with an automatic sentence alignment model that was trained on MC-CLEAN. MC-NOISY is sourced from around 7.8K medical reviews, with ∼100K sentence pairs across languages.

MULTICOCHRANE enables systematic evaluations of medical text simplification models. Here we evaluate a range of large pre-trained language models in both zero-shot and fine-tuned settings on the task of text simplification across four languages. In addition to automatic evaluations, we report human assessments covering simplicity, fluency and factuality (Devaraj et al., 2022). We also report the correlation between automatic metrics and these human assessments, which we find to be mostly weak. Our results show that while pre-trained models are effective at simplification in English, their abilities degrade significantly on other languages,

---

to overcome this limitation, it is not ideal due to variability in translation quality across languages (Ruder et al., 2021) and other issues such as cost and interpretability as discussed in Vu et al. (2022). This paper evaluates a simplify-then-translate pipeline.

[2]https://www.cochranelibrary.com

where they tend to introduce factuality errors. GPT-3 (davinci; zero-shot) yields outputs that are comparatively factually accurate, but which tend to be extractive and so not adequately simplified. Outputs from Flan-T5 (Base; zero-shot) significantly degrade in quality when targeting languages other than English, producing many factual and fluency errors. We also analyze the approach of translating English simplifications to other languages, which in many instances is able bypass these issues.

We publicly release MULTICOCHRANE, model outputs, and all human judgments collected for this work (https://github.com/SebaJoe/MultiCochrane), hoping to motivate future work on multilingual medical text simplification to advance model performance across languages.

## 2 Related Work

The largest resources used to train automatic simplification models are two general domain corpora: the Wikipedia-Simple Wikipedia corpus (Zhu et al., 2010; Woodsend and Lapata, 2011; Coster and Kauchak, 2011; Jiang et al., 2020), and the Newsela corpus (Xu et al., 2015; Jiang et al., 2020). This paper focuses on simplification in the medical domain, which is important if we are to bridge the information gap exemplified by low medical literacy levels worldwide (Kickbusch et al., 2013).

**Medical Simplification Datasets.** While there have recently been increasing efforts to create parallel corpora for medical text simplification in English (Van den Bercken et al., 2019; Cao et al., 2020; Devaraj et al., 2021; Guo et al., 2022; Basu et al., 2023), data in other languages remain scarce. Grabar and Cardon (2018) constructed the CLEAR dataset in French, part of which is derived from 13 Cochrane articles; Trienes et al. (2022) introduced a dataset on German consisting of clinical notes. Other prior work in non-English medical text simplification has focused on lexical simplification (Abrahamsson et al., 2014; Kloehn et al., 2018; Alfano et al., 2020), where the primary focus was on substituting synonyms and semantically related terms. In terms of data sourcing, our work is most similar to Devaraj et al. (2021) (which is in English only). However, their work derived roughly-aligned paragraphs consisting of specific sections (methods and conclusions only) of the Cochrane abstracts and PLS; by contrast, we derive manual and automatically aligned sentences for full abstracts.

**Multilingual Simplification Datasets.** Multilingual datasets have been introduced for the task of summarization. The MLSUM dataset (Scialom et al., 2020) is a large summarization corpus containing article/summary pairs in five languages, sourced from newspaper articles. General-domain non-English datasets for simplification also exist as extensively surveyed by Ryan et al. (2023). Notably, MUSS (Martin et al., 2022) used Common Crawl to mine millions of sequence pairs in English, Spanish, and French. The significant difference between MUSS and our dataset is MUSS's lack of cross-lingual alignment, i.e., alignment of sequence pairs from one language to another. Agrawal and Carpuat (2019) derived noisily aligned English-Spanish data from the Newsela corpus (Xu et al., 2015). However, to date, there is no aligned dataset for text simplification where one source sentence is paired with target sentences in multiple languages simultaneously as in our work.

## 3 The MULTICOCHRANE Corpus

We source MULTICOCHRANE from publicly available technical abstracts and plain-language summaries of Cochrane systematic reviews; these are comparable texts, though not parallel (Devaraj et al., 2021). These abstracts and PLS exist in several languages (detailed in Sections 3.1.2 and 3.2), aligned almost one-to-one with their English counterparts. This allows annotations and alignments to be easily adapted to create a multilingual dataset. In total, we use 7,755 abstract-PLS pairs from the Cochrane Library. We first create sentence-aligned MC-CLEAN (Section 3.1) of 101 abstract-PLS pairs using a mixture of manual alignments for English and semi-automatic alignments with partial verification for other languages. The rest of the abstracts were used for the creation of the large-scale noisy MC-NOISY (Section 3.2) by automatic alignments and filtering. The total number of sentences in all subsets are shown in Table 1.

### 3.1 Clean Alignments: MC-CLEAN

For MC-CLEAN, we first create a manually aligned subset of 101 abstracts for English. We then automatically align for other languages by exploiting Cochrane's natural 1-1 sentence alignment from English to other languages; this alignment was verified by annotators on a sizable sample of the data. Our annotation team consists of 5 trained linguistics students with native English proficiency and

| Dataset | en→en | en→es | en→fr | en→fa |
|---------|-------|-------|-------|-------|
| MC-NOISY | 60,058 | 58,235 | 34,604 | 30,822 |
| + filtering | 29,703 | 29,025 | 17,033 | 15,389 |
| MC-CLEAN | 1,632 | 1,602 | 1,111 | 646 |
| training set | 1,136 | 1,084 | 798 | 424 |
| + filtering | 329 | 326 | 227 | 125 |

Table 1: Statistics of MULTICOCHRANE corpus for English (en), Spanish (es), French (fr), and Farsi (fa). MC-NOISY is only used for training. Filtering (Section 4) selects the most similar pairs of aligned sentences at the cost of higher extractiveness.

native or advanced mastery over one or more of the other languages covered in this work. These annotators do not have medical backgrounds, but they do have a strong background in research and in evaluating technical texts. We consider this to be sufficient enough to complete the tasks described in this paper.

#### 3.1.1 English Annotation

Annotators were provided with abstract-PLS pairs and were tasked with aligning sentences in the abstract with sentences in PLS according to their similarity in content. This is a challenging task: annotators had to be able to understand complex medical texts, and align sentences by repeatedly searching through text on both the simple and complex sides. To assist with their annotation, we built the ANNO-VIEWER tool inspired by Jiang et al. (2020) to enable annotators to efficiently align sentences (see screenshots in Appendix J).[3] This annotation tool also has functionality allowing the annotator to use existing semantic similarity measures (listed in Section 3.2) to help find sentence alignments faster. When a sentence is selected for alignment, the tool can sort the other document (either abstract or PLS) by which sentences are most similar to the selected sentence. After aggregating all alignments for a document into a bipartite graph for analysis, we processed the data into individual sentence pairs (see Appendix D for more details).

Due to the high cognitive load imposed by aligning medical documents, we first selected a set of 25 documents aligned by at least 2 annotators (15 of which aligned by at least 3 annotators). Once we were confident of their inter-annotator agreement (see "Inter-annotator Agreement" below), the rest of the 75 documents were then single-annotated.

---

[3] We release the ANNO-VIEWER annotation tool publicly together with our datasets.

| | |
|---|---|
| Aligned Complex Sentences | 1,325 |
| Aligned Simple Sentences | 1,165 |
| Unaligned Complex Sentences (Deletion) | 1,966 |
| Unaligned Simple Sentences (Elaboration) | 591 |
| Average Deletion Ratio | 60.1% |
| Average Elaboration Ratio | 28.2% |
| Token Compression (complex→simple) | 1.14 |

Table 2: Statistics on unaligned and aligned sentences for the English portion of MC-CLEAN.

**Inter-annotator Agreement.** We used the aforementioned 25 articles to calculate inter-annotator agreement based on the F1 score, following prior research on word and sentence alignment (Jiang et al., 2020; Lan et al., 2021).[4] For each annotator, their alignments were evaluated against the majority vote of alignments from other annotators.

Overall, the macro-average F1 score across annotators was 0.89, indicating high agreement for the alignment task. Non-overlapping alignments are rare; a major source of disagreement between annotators was related to multiple alignments made to the same simple sentence (i.e., *multi-alignments*) that were either captured by some annotators but excluded by others. Since the content in the abstracts is highly technical and in some cases contrasted significantly in style and vocabulary with the plain-language summary, annotators had difficulties in determining whether some sentences conveyed similar meaning. Nevertheless, annotators were in general conservative, only aligning sentences when they were confident in the alignments. Therefore, there is a low prevalence of incorrect alignments. The overall alignments are high-precision. Annotators typically used ANNO-VIEWER's sentence similarity sorting function to verify their alignments or to identify additional multi-alignments.

**Dataset Summary.** Table 2 presents additional statistics on the aligned and unaligned sentences in both complex and simple texts in the English portion of MC-NOISY. For complex texts (i.e., technical abstracts), an unaligned sentence indicates that its information is not present (i.e., *deleted*) in the simplified texts (i.e., plain-language summaries). On average, just over 60% of sentences in complex texts are deleted, indicating that most of the core information from complex texts are retained

in simple texts. In PLS, an unaligned sentence indicates added information not present in the complex version. We observe that an overwhelming majority of added information *elaborates* on concepts or defines terms (Srikanth and Li, 2021). The average elaboration ratio (the ratio of unaligned to total sentences in a PLS) is less than half of the average deletion ratio (ratio of unaligned to total sentences in a technical abstract), indicating that simplified texts tend to be mostly concise with this core information. The average token compression ratio (ratio of simple to complex tokens) show the presence of longer simplified sentences.

### 3.1.2 Multilingual Data

The Cochrane Library provides abstracts and PLS in multiple languages via translations that have been produced through a combination of volunteer and professional work, as well as human-verified machine translation (Ried, 2023). To create the multilingual portion of our dataset, i.e., pairs of semantically equivalent *(source, target)* sentences where *source = English abstract, target ∈ PLS in {en, es, fr, ...}*, we use the English PLS as a bridge to align the English abstract with PLS in other languages. The PLS of non-English languages mostly correspond 1-1 sentence-wise to the English PLS.

We use a sentence aligner that combines LASER multilingual sentence embeddings (Artetxe and Schwenk, 2019) with a dynamic programming-based alignment algorithm (Thompson and Koehn, 2019) to align sentences in the English versions to those in other languages. We did this for the entire set of 7,755 abstract/PLS pairs. Multilingual sentence pairs in MC-CLEAN consist of these alignments that belong to the 101 articles manually aligned for English. The other 7,654 articles were used to create MC-NOISY (Section 3.2).

**Human Verification.** To verify that the multilingual sentence alignments across different languages of the PLS are valid, we asked 3 of our bilingual annotators to evaluate a random sample of 400 sentence pairs to verify that each alignment is a direct translation. Each annotator was highly proficient in both the languages of the alignments being verified. Annotators found zero instances of misalignment. We also analyzed the number of sentences of the English abstract/PLS and their multilingual counterparts to find any instances where is a huge mismatch. We did not find any such instances. The data did include instances where

---

[4]Alignment requires $N$ (sentences) by $M$ (sentences) comparisons, while only a small amount of which being positive (i.e., aligned). Thus F1 is more commonly used than correlation measurements for calculating agreement for this task.

one or two sentences in the article may have been split or combined in their multilingual versions. The DP-based alignment algorithm that was used took care of these exceptions. These assessments, combined with information about the Cochrane Library's methodology in making these translations, instills high confidence that the derived multilingual sentence alignments are valid.

**Dataset Summary.** While Cochrane provides a large amount of diverse multilingual data, every article is not mapped to every available language. Consequently, the number of articles available for each language varies. The distribution of these articles across language is shown in Figure 2.

For this work, we selected the top three most frequent non-English languages — Spanish, French, and Farsi — to use for training and evaluation in Section 5. We report the resulting number of paired sentences in each language in Table 1.

### 3.2 Fully Automated Alignments: MC-NOISY

While MC-CLEAN provides clean and accurate alignments across multiple languages, human alignment of medical documents is challenging to scale. Fortunately, MC-CLEAN enables us to accurately evaluate a variety of *automatic* sentence alignment methods; a good automatic alignment method will help create silver-standard alignments expanding over the entirety of the Cochrane dataset. This is especially important for multilingual medical simplification, where the number of annotated alignments is significantly lower for non-English languages.

**Automatic Alignment on English Data** The automatic alignments for MC-NOISY were derived using a BERT-based CRF alignment model (Jiang et al., 2020). This method outperformed other alignment methods when evaluated on a subset of MC-CLEAN. Further details about these experiments can be found in Appendix E.

**Multilinguality** We use the same 1-1 alignment property between English and other languages in Cochrane abstracts and PLS described above (Section 3.1.2) to derive the multilingual portion of MC-NOISY from the English data described in Section 3.2. As shown in Figure 2, the distribution of the articles and sentence pairs across language in the noisy dataset is similar to that of the human-annotated dataset.

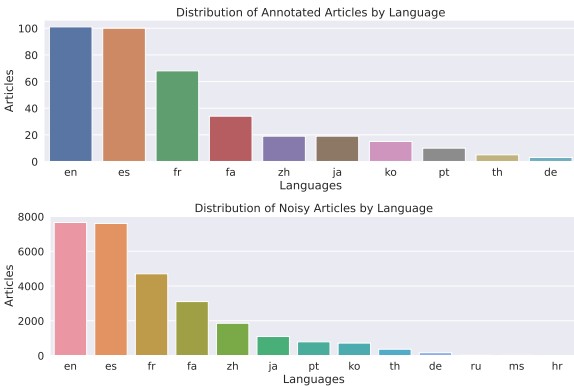

Figure 2: For English abstracts included in MC-CLEAN and MC-NOISY, the number of corresponding PLS across languages. From left to right: English (en), Spanish (es), French (fr), Farsi (fa), Chinese (zh), Japanese (ja), Portuguese (pt), Korean (ko), Thai (th), German (de), Russian (ru), Malay (ms), and Croatian (hr).

## 4 Filtering

Although the CRF model handles unaligned sentences by default, the resulting data remains noisy. We therefore adapt the method described in Kim et al. (2021), initially used in the context of sentence-splitting, to further filter misalignments in MC-NOISY. Ultimately, this method was applied to the entire training set, which includes the entirety of MC-NOISY as well as the portion of MC-CLEAN not used for testing or validation. This method can be described as follows: For each sentence pairing in the training set, we derive the set of lemmatized tokens in the complex and simple sentence denoted by $\mathcal{L}_c$ and $\mathcal{L}_s$ respectively. If the proportion of the overlapping tokens from the simplified sentence exceeds a threshold $r$, it is included in the filtered dataset: $r = |\mathcal{L}_c \cap \mathcal{L}_s|/|\mathcal{L}_s|$.

We used $r = 0.5$ to strike a balance between the number of potential misalignments in the dataset and the extractiveness of the data. Overall, as evident in Table 1, this filtering process reduced the dataset size by about half. Filtering reduced the MC-CLEAN portion by a larger proportion, indicating that human data is less lexically similar as opposed the noisy data.

## 5 Evaluation of Simplification Models

MULTICOCHRANE enables extensive evaluation of multilingual sentence simplification systems simultaneously across 4 languages. In this section we present the first study of this kind (to our knowledge), assessing zero-shot and fine-tuned models (Section 5.1) with both automatic metrics (Sec-

tion 5.2) and human evaluation (Section 5.3) for medical text simplification.

## 5.1 Models

We experiment with zero-shot and fine-tuned models, as well as a simplify-then-translate pipeline.

**Zero-shot models:**

- **GPT-3 zero-shot**. we evaluate zero-shot simplifications generated by the GPT-3-davinci-003 model using a prompt similar to that used in August et al. (2022), adapted for multilinguality:

  *My fifth grader asked me what this sentence means: [TEXT TO SIMPLIFY] I rephrased it for him in [LANG], in plain language a fifth grader can understand:*

- **GPT-3 simplify-then-translate**. GPT-3 has been shown to have strong performance on English simplification for medical texts (Shaib et al., 2023). Thus, we evaluate a pipeline model first using GPT-3 for English simplification using the above prompt, and then translating the simplified text to other languages with a strong translation model (Google's Cloud Translation API).

- **Flan-T5 zero-shot**. We also evaluate zero-shot Flan-T5$_{base}$ (Chung et al., 2022) performance. We used a simple prompt, specifically prepending *"Simplify this sentence:"* to the input (complex) sentence.[5] When simplifying to languages other than English, the prompt is changed to *"Simplify this sentence in [LANG]:"*. Unfortunately, we were unable to generate simplifications in Farsi using Flan-T5, so we only evaluated this system on English, Spanish, and French.

**Fine-tuned models:**

- **mT5** We fine-tune separate mT5$_{base}$ models (Xue et al., 2021) on different language pairs: (English, English), (English, Spanish), (English, French), and (English, Farsi).

- **Flan-T5 fine-tuned** We further evaluate fine-tuned versions of Flan-T5$_{base}$ for each language pair, following the setup of its zero-shot counterpart. This system also failed to generate Farsi outputs.

---

[5]Preliminary experiments showed that August et al. (2022)'s prompt, though worked well on GPT-3, did not yield good results with Flan-T5.

|    | train | test | validation |
|----|-------|------|-----------|
| **en** | 7728 (61194/30032) | 22 (395) | 5 (83) |
| **es** | 7643 (59319/29351) | 22 (395) | 5 (80) |
| **fr** | 4709 (35402/17260) | 11 (214) | 4 (65) |
| **fa** | 3117 (31246/15514) | 8 (148) | 3 (56) |

Table 3: Train/test/validation splits used for evaluation. Both the number of articles and the total number of alignments are displayed (for training: unfiltered ($r = 0$)/filtered ($r = 0.5$)).

| | System | BLEU | BS | SARI | LCS |
|---|--------|------|-----|------|-----|
| **English** | GPT3$_{zero}$ | 2.38 | 0.8774 | 42.111 | 0.569 |
| | Flan-T5$_{zero}$ | 8.12 | 0.8810 | 39.057 | 0.340 |
| | mT5$_{r=0}$ | 7.66 | 0.8785 | 40.219 | 0.466 |
| | mT5$_{r=0.5}$ | 8.82 | 0.8843 | 39.579 | 0.379 |
| | Flan-T5$_{r=0.5}$ | 8.70 | 0.8875 | 39.526 | 0.319 |
| **Spanish** | GPT3$_{zero}$ | 5.70 | 0.7412 | 37.972 | 0.361 |
| | GPT3$_{trans}$ | 3.11 | 0.7188 | 41.123 | 0.561 |
| | Flan-T5$_{zero}$ | 3.04 | 0.7100 | 37.890 | 0.367 |
| | mT5$_{r=0}$ | 4.97 | 0.7337 | 41.224 | 0.564 |
| | mT5$_{r=0.5}$ | 5.18 | 0.7381 | 39.522 | 0.508 |
| | Flan-T5$_{r=0.5}$ | 5.21 | 0.7376 | 40.064 | 0.486 |
| **French** | GPT3$_{zero}$ | 4.42 | 0.7325 | 38.096 | 0.380 |
| | GPT3$_{trans}$ | 2.82 | 0.7134 | 40.941 | 0.579 |
| | Flan-T5$_{zero}$ | 1.63 | 0.6989 | 39.452 | 0.449 |
| | mT5$_{r=0}$ | 2.35 | 0.7146 | 40.412 | 0.602 |
| | mT5$_{r=0.5}$ | 2.96 | 0.7157 | 39.096 | 0.553 |
| | Flan-T5$_{r=0.5}$ | 3.89 | 0.7284 | 40.989 | 0.528 |
| **Farsi** | GPT3$_{zero}$ | 1.36 | 0.7103 | 41.080 | 0.497 |
| | GPT3$_{trans}$ | 1.16 | 0.7094 | 43.738 | 0.577 |
| | mT5$_{r=0}$ | 2.21 | 0.7111 | 43.280 | 0.622 |
| | mT5$_{r=0.5}$ | 2.67 | 0.7139 | 43.154 | 0.631 |

Table 4: Automatic metrics for all generations.

**Training Setup.** We evaluate models on train/test/validation splits shown in Table 3. The train split is composed of both the entirety of MC-NOISY and a portion of MC-CLEAN while test and validation splits are subsets of MC-CLEAN. Fine-tuned models (mT5, Flan-T5) were trained over a single epoch with a learning rate of 5e-5 and used the AdamW optimizer. For both mT5 and Flan-T5 (zero-shot and fine-tuned), nucleus sampling with a top-p of 0.95 was used as the decoding strategy. GPT-3 was evaluated using a temperature of 0.7.

## 5.2 Automatic Evaluation

**Metrics.** Model outputs were evaluated through five automatic evaluation metrics: BLEU (Post, 2018), BERTScore (Zhang et al., 2020), SARI (Xu et al., 2016), and the Least-Common Subsequence distance metric (LCS) (Bakkelund, 2009) This dis-

tance metric has an inverse relationship with the actual least common subsequence (lower means more extractive).

We do not include Flesch-Kincaid as a metric, for reasons stated in Tanprasert and Kauchak (2021), which provides a compelling analysis on how easily it can be misinterpreted, and instead opt for human evaluation of simplicity in Section 5.3.

**Results.** Results of these evaluations are presented in Table 4. The data in this table show a common trend in that filtering (Section 3.2) largely improves performance, indicating that filters improve data quality. A very visible trend in this data, is that across the board, with the exception of the GPT results, English simplifications vastly outperform those of other languages. This probably due to the vast resource gap between English and the other languages, as well a possible bias towards English in the pre-trained mT5-base model.

We observe exceedingly low BLEU scores when the simplified version is used as reference, compared to the complex version. Since BLEU is lexical-centric, this reveals that models tend to translate rather than simplify. Simplified sentences in the Cochrane dataset frequently featured insertions and elaborations, creating at times an information mismatch between the model outputs and the corresponding reference simplifications. Moreover, the style in which PLS were written varied greatly (Devaraj et al., 2021).

Another attribute that is of importance when judging simplification is the level of extractiveness, or how much the output copied the input. Based on the LCS metric, it seems that while GPT-3 produces less extractive outputs than English mT5, for the other languages the opposite is true. We discuss this further in Section 5.3.

Extractiveness and poor multilingual simplifications are also evident in the results for zero-shot Flan-T5 generations. For multilingual simplifications, in particular, fine-tuning significantly increased the LCS distance while improving BLEU and BERTScore metrics, indicating higher quality and less extractive simplifications. Interestingly, English simplifications generated by both settings have a significantly lower LCS distance compared those generated by other systems. Why this occurs is unclear to us and requires further analysis.

The simplify-translate approach to simplification in different languages elicited similar results as its English counterpart and clearly produces less

| | System | Fact. | Fluency | Simpl. |
|---|---|---|---|---|
| English | mT5$_{r=0}$ | 0.87 | 0.50 | 0.24 |
| | mT5$_{r=0.5}$ | 0.59 | 0.58 | 0.31 |
| | Flan-T5$_{r=0.5}$ | 0.50 | 0.25 | 0.20 |
| | GPT3$_{zero}$ | **0.33** | 0.25 | **0.02** |
| | Flan-T5$_{zero}$ | 0.39 | **0.18** | 0.18 |
| | Reference | 0.25 | 0.07 | 0.02 |
| Spanish | mT5$_{r=0}$ | 1.05 | 0.44 | 0.50 |
| | mT5$_{r=0.5}$ | 0.89 | 0.46 | 0.41 |
| | Flan-T5$_{r=0.5}$ | 0.76 | 0.83 | **-0.02** |
| | GPT3$_{zero}$ | **0.02** | **0.00** | 0.90 |
| | GPT3$_{trans}$ | 0.44 | 0.02 | 0.51 |
| | Flan-T5$_{zero}$ | 1.22 | 1.05 | 0.78 |
| | Reference | 0.50 | 0.02 | 0.17 |
| French | mT5$_{r=0}$ | 1.25 | 0.67 | -0.40 |
| | mT5$_{r=0.5}$ | 1.3 | 0.70 | -0.03 |
| | Flan-T5$_{r=0.5}$ | 0.97 | 0.63 | 0.04 |
| | GPT3$_{zero}$ | 0.43 | **0.27** | 0.51 |
| | GPT3$_{trans}$ | **0.38** | 0.44 | **0.01** |
| | Flan-T5$_{zero}$ | 1.11 | 1.32 | 0.49 |
| | Reference | 0.09 | 0.04 | -0.03 |
| Farsi | mT5$_{r=0}$ | 0.65 | 0.49 | -0.41 |
| | mT5$_{r=0.5}$ | 0.65 | 0.59 | -0.53 |
| | GPT3$_{zero}$ | 0.40 | 0.14 | **-0.28** |
| | GPT3$_{trans}$ | **0.11** | **0.06** | -0.36 |
| | Reference | 0.87 | 0.82 | -0.51 |

Table 5: Summary of human evaluation results across all evaluated systems as well as the reference simplification, showing average factuality (0-2, lower=better), fluency (0-2, lower=better) and simplicity (-2 (oversimplify)–2 (too hard)) ratings. Best system performance bolded.

extractive generations compared to zero-shot multilingual simplification. In terms of LCS and SARI, this approach resulted in similar scores as those produced by fine-tuned models. While it is appears that the simplify-translate approach improved upon the zero-shot multilingual performance for GPT, it is difficult to draw conclusions about its performance relative to fine-tuned models using automatic metrics alone.

## 5.3 Human Evaluation

We perform human evaluation of the factuality, linguistic quality, and simplicity of model outputs. We also evaluate the **reference simplifications**. 100 outputs were evaluated for each system. While the alignment may be sound, the references may still contain excess deletions or inconsistencies with the original text; prior work Devaraj et al. (2022) found similar phenomena with the Newsela and Wiki datasets. Annotators were blinded to the sys-

tem ID (or whether they are evaluating the manually aligned pairs from the texts themselves).

**Metrics.** (1) **Factual faithfulness**: We measure overall factuality using a 0-2 scale, where 0 indicates no factual issues and 2 indicates severe factual issues. (2) **Fluency**: We rate linguistic fluency of outputs using a scale of 0-2, where 0 indicates fluent output and 2 indicates severe linguistic issues. (3) **Simplicity**: Simplicity is assessed on a scale of -2 to 2, where -2 indicates heavy oversimplification, 0 indicates ideal simplification, and 2 indicates heavy under-simplification. The specific criteria are further described in Appendix A.

**Results.** The human evaluation results presented in Table 5 agree with many of the automatic metrics. First, annotators found English outputs from the mT5 models to be more factual and fluent than those from the other languages. Similarly, factuality and fluency also improved with the filtered dataset for the most part (as was also indicated by the automatic metrics). This is probably due to filtering removing less extractive sentence pairs from the training set. However, for French and Farsi, filtering slightly worsened the factuality and fluency, perhaps due to the fewer number of examples in the data. For simplicity, with the exception of English, filtering also seems to make outputs simpler.

There is a stark difference between the GPT-3 and the mT5 outputs. GPT-3 produced significantly more faithful and fluent text while mT5 outputs were deemed simpler with the exception of English. However, qualitative feedback from annotators suggests that non-English GPT-3 outputs are skewed due to a severe degree of extractiveness; in some cases, the output is simply a word-for-word translation of the input. This inflates the factuality and fluency scores, but this level of extractiveness is manifest in the simplicity scores; GPT-3 does proper simplification only for the English to English case.

The trends in the results for Flan-T5 mirror those for GPT-3. When comparing zero-shot generations to fine-tuned generations, it is clear that Flan-T5 is limited in its ability to perform multilingual simplification. Not only are generations less simplified, but they frequently have factual and fluency errors.

The simplify-translate approach proved to be fairly strong compared to other systems which directly performed multilingual simplification. The current state of direct multilingual simplification

in these large language models seems to be not yet surpassing monolingual simplification combined with machine translation. However, as discussed in Section 1, a pipeline system does come with its drawbacks in terms of cost and robustness.

Annotators also evaluated the reference labels of each complex input from the dataset. Overall, with the exception of Farsi, these references were deemed relatively fluent and simplified. However, due to insertions and deletions of information that naturally occur in the data, factuality errors were detected by annotators; these findings echo those in Devaraj et al. (2022).

**Examples of common errors.** One common error that was found in outputs from all evaluated systems were mistranslated acronyms. Medical acronyms often differ from one language to another. However, many systems are limited in their ability to properly translate this technical knowledge. Usually, the acronym is either not translated, or translated incorrectly, often randomly. The following example in French was generated by mT5 fine-tuned on the unfiltered version of MC-NOISY.

*Input complex sentence*: Stress and a sedentary lifestyle are major determinants of cardiovascular disease (**CVD**).
*Output*: Le stress et le règlement inactif sont des causes importantes de maladies cardiovasculaires (**AVC**) les plus courants.
*Gloss*: The stress and inactive regulation are the most common important causes of cardiovascular disease (AVC)

The correct translation of CVD, or cardiovascular disease, in French is MCV, not AVC (French acronym for stroke).

Fluency issues were especially common in outputs generated by mT5 and Flan-T5. In the case of Flan-T5, in particular, misspellings and incorrect applications of gendered language are quite prevalent. Consider the following example in Spanish generated by Flan-T5:

*Input complex sentence*: The rates were 24.5 for the sponge and 10.9 for the diaphragm in the UK trial.
*Output*: Las tasas de asistencia respiratoria fueron de un 2,4 para la secreción **del** vejiga y 10,9 para el **diafragmata**.
*Gloss*: The rates of respiratory assistance were 2.4 for the secretion of the bladder and 10.9 for the diaphragm.

This example has a number of errors. There are many hallucinations that include the insertion and substitution of irrelevant information. However, with respect to fluency, the main issue is the incorrect gender used for the word "vejiga" as well as the misspelling of the word "diafragma" as "diafragmata". The important thing to note here is that

most of these errors are localized to technical words or phrases. More work needs to be done to adapt large language models to these niche vocabularies.

## 6  Conclusions

This paper introduces the first human-aligned multilingual text simplification dataset, MC-CLEAN, for the medical domain for the languages of English, Spanish, French, and Farsi. We further expanded this dataset using automatic alignments to create a much more vast but noisier dataset, MC-NOISY, that could be used for training language models. We also performed an evaluation of multilingual simplification for various systems, testing some of their zero-shot capabilities as well their performance when fine-tuned on MC-NOISY.

## Limitations

MULTICOCHRANE has a few limitations. Firstly, for a given article, there does not exist a consistent set of multilingual versions for that article. As exemplified in Figure 2, the number of articles in which there exists a multilingual version varies depending on the language. This uneven distribution does favor more well resourced languages, with Farsi being a notable exception.

Looking at the wider scope of text simplification, sentence-level text simplification does have a few drawbacks. Especially within the medical domain, contextual information is helpful for generating good simplifications. In MC-CLEAN elaborative information was often left unaligned. Simplifying at a paragraph-level or a document-level could have made use of this additional data to generate more useful simplifications.

Because of limitations with the amount of computing capability and time, the largest models available for Flan-T5 and mT5 were not used for evaluation. The models that we used were the best available models that we could have evaluated within our reasonable constraints.

## Ethical Concerns

This work does not use any full texts of Cochrane reviews. The research use of publicly available abstracts, including both technical abstracts and multi-lingual plain language summaries, is considered fair use. Note that the Cochrane dataset, in similar capacity as ours, has already been released publicly by prior work (Devaraj et al., 2021; Guo et al., 2020).

Regarding compensation for annotators, all the annotators were paid a wage of $15 for every hour of work.

We recognize the dual use of language models and the possibility of factual errors and hallucinations in their generations, and we believe that this line of work is necessary both to advance technology for social good while also acknowledging these risks. The inaccessibility of medical information and health illiteracy are some of the leading reasons for real health consequences including more hospital admissions, emergency room visits, and poorer overall health (Berkman et al., 2011). Making medical information accessible is one of the best ways to tackle health literacy, and that is the core of what a simplification system is aimed to do. Additionally, medical misinformation is one of the most series issues highlighted in the COVID-19 pandemic; one contributing reason for this is the lack of health literacy among the general public. By simplifying trustworthy evidence, we hope to empower the public with a keener eye for such misinformation. Factual errors is one of the key aspects studied in this work; we perform a thorough evaluation dissecting issues that can come from these models, especially in a multilingual setting. We believe rigorous evaluation, as done in this work, is one of our best tools to demystify language models, and help the community understand the issues at hand. With such understanding, we hope to point to future directions that collectively, we will be able to provide factual, readable, multilingual access to medical texts.

## Acknowledgements

We acknowledge Pouya Nekouei for his help with the Farsi language. This research was partially supported by National Science Foundation (NSF) grants IIS-2145479, IIS-2144493 and IIS-2112633, and by the National Institutes of Health (NIH) under the National Library of Medicine (NLM) grant 2R01LM012086.

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

# A Human Evaluation Framework

The following is the exact framework that annotators used for evaluating simplification outputs.

| **Factual (faithful to input)** | **Choices - 0-2**
Give an overall rating on how factually consistent the model-generated output is on a scale from 0 to 2, where 0 is completely factual, 1 indicates inconsequential factual errors, and 2 indicates severe factual errors.

*Example:*
*Rating 0:*
*Original: We discovered a high percentage of study participants experienced adverse reactions while being treated for ischemic heart disease.*
*Output: We found that many study participants had bad side effects from heart disease treatments.*

*Rating 1:*
*Original: We discovered a high percentage of study participants experienced adverse reactions while being treated for ischemic heart disease.*
*Output: We discovered some participants had bad side effects from heart disease treatments.*

*Rating 2:*
*Original: We discovered a high percentage of study participants experienced adverse reactions while being treated for ischemic heart disease.*
*Output: We discovered participants had no side effects from heart disease treatments.* |
|---|---|
| **Fluency** | **Choices - 0-2**
Give an overall rating on how well the model-generated output follows grammatical rules and appears as a natural sentence, where 0 indicates no fluency issues, 1 indicates superficial issues, and 2 indicates severe fluency issues that obfuscate the meaning of the sentence.

*Example:*
*Rating 1:*
*Original: We discovered a high percentage of study participants experienced adverse reactions while being treated for ischemic heart disease.*
*Output: We found that many study participants had badly side effects from heart disease treatments.*

*Rating 2:*
*Original: We discovered a high percentage of study participants experienced adverse reactions while being treated for ischemic heart disease.*
*Output: In heart disease treatments, many participants had effects bad on the side.* |

| Simplicity | **Choices - (-2 - 2)** |
| --- | --- |
| | Rate the level of simplification that occurred from the original English sentence to the model-generated output. Note: since the input readability is varied, please make judgements independent of the input. |
| | **-2:** Output is severely oversimplified to the point where the original intent of the original sentence has been lost. |
| | **-1:** Output is oversimplified, missing some important details. However, the intent of the original sentence is preserved. |
| | **0:** Output is ideally simplified. It is readable to the average layman and does not omit important details. |
| | **1:** Output should be simplified more and include uncommon technical terms without any elaboration/explanation. |
| | **2:** Output should be simplified MUCH more. There is little to no change in the style of the sentence from the original sentence, and the output is difficult for the average layman to understand. |
| | |
| | *Example:* |
| | *Rating -2:* |
| | *Original: We discovered a high percentage of study participants experienced adverse reactions while being treated for ischemic heart disease.* |
| | *Output: Study participants had some effects while being treated for some disease.* |
| | |
| | *Rating -1:* |
| | *Original: We discovered a high percentage of study participants experienced adverse reactions while being treated for ischemic heart disease.* |
| | *Output: Study participants had bad effects while being treated for heart disease.* |
| | |
| | *Rating 0:* |
| | *Original: We discovered a high percentage of study participants experienced adverse reactions while being treated for ischemic heart disease.* |
| | *Output: We found that many study participants had bad side effects from heart disease treatments.* |
| | |
| | *Rating 1:* |
| | *Original: We discovered a high percentage of study participants experienced adverse reactions while being treated for ischemic heart disease.* |
| | *Output: We found that many study participants had experienced adverse effects from ischemic heart disease treatments.* |
| | |
| | *Rating 2:* |
| | *Original: We discovered a high percentage of study participants experienced adverse reactions while being treated for ischemic heart disease.* |
| | *Output: We discovered a high percentage of study participants experienced antagonistic reactions while being remedied for ischemic heart disease.* |

| Factuality | Fluency | Simplicity | Combined |
|------------|---------|------------|----------|
| 0.362 | 0.549 | 0.372 | 0.464 |

Table 7: Randolph's kappa on English outputs evaluated for agreement.

| Alignment Type | Method A | Method B |
|----------------|----------|----------|
| 1 - 1 | 36.4% | 50.3% |
| N - 1 | 20.3% | 36.6% |
| 1 - N | 9.4% | 10.1% |
| N - M | 33.9% | 2.9% |

Table 8: Distribution of alignment types in MC-CLEAN for each method used.

## B Human Evaluation Score Distributions

Figure 3 show the distributions of the human evaluation scores for all evaluated systems.

## C Human Evaluation Agreement

In addition to evaluating outputs in their respective languages, all human evaluators also evaluated a set of 100 examples in English. These were collected to estimate how similarly annotators would evaluate outputs.

To analyze agreement, we use Randolph's kappa (Randolph, 2005), a free-marginal version of Fleiss' kappa. These values are presented in Table 7 for the different human evaluation criteria along with one for all of them combined. These results show there is moderate agreement among annotators.

## D Alignment Analysis

Since annotators were given total freedom to align sentences, many different alignment relationships are present in MC-CLEAN. To quantify the proportion of alignments that belong to these alignment relationships, two methods were used for analysis. For future reference, 1-1, N-1, 1-N, and N-M alignments refer to number of complex sentence aligned to the number of simple sentences in an alignment group.

**Method A.** This method treats alignments in a document as edges in a bipartite graph, with complex and simple sentences as vertices. Relationships are found by tracking the connected components in this graph through a depth-first search. The type of relationship is determined by a 2-tuple of the number of complex sentences and the number of simple sentences in a connected component.

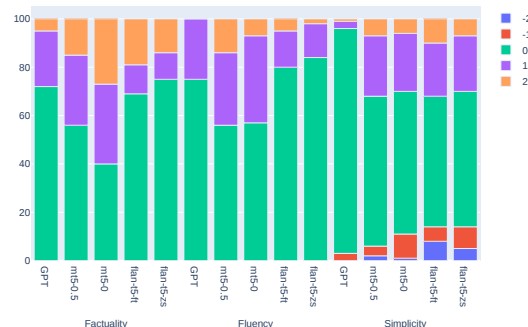

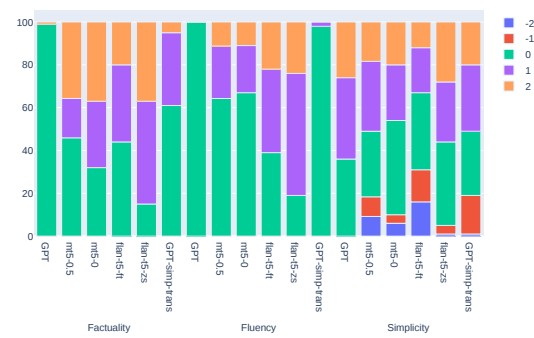

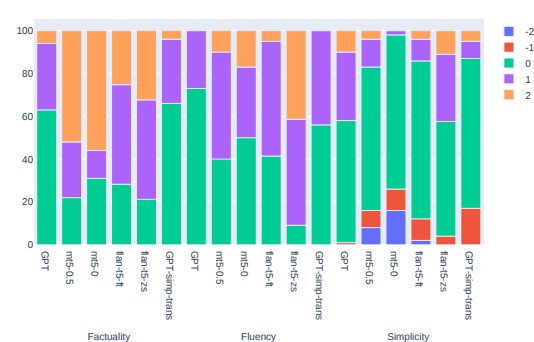

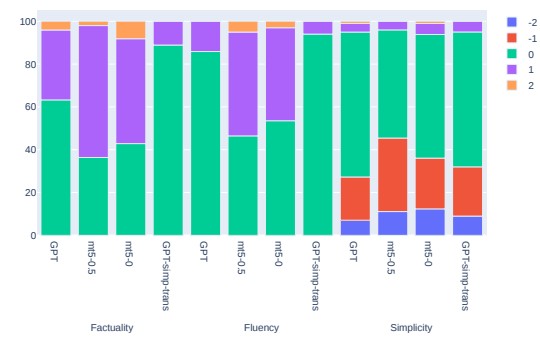

Figure 3: These graphs show the distribution of human evaluation scores for each evaluated system in English, Spanish, French, and Farsi.

| Method | Precision | Recall | F1 |
|---|---|---|---|
| Jaccard | 0.737 | 0.238 | 0.360 |
| Sentence-BERT | 0.437 | 0.416 | 0.426 |
| TFIDF | 0.372 | 0.354 | 0.363 |
| BERTScore | 0.468 | 0.446 | 0.457 |
| Neural-CRF$_{Newsela}$ | 0.454 | **0.592** | 0.514 |
| Neural-CRF$_{Cochrane}$ | **0.781** | 0.466 | **0.584** |

Table 9: Precision, Recall, F1 for sentence alignment methods evaluated on the manually aligned English portion of MC-CLEAN.

**Method B.** This method is an extension of Method A that specifically targets on breaking up N-M alignment relationships. While Method A counts any connected component as an alignment group, this method requires that each component must be fully connected (every complex sentence is aligned to every simple sentence in the group) as well. As such, an N-M alignment group that is not fully connected is broken up into smaller fully connected components.

The results of this analysis is presented in Table 8. The results from Method A can be viewed as a lower bound for 1-1, N-1, and 1-N, alignment types. Method B could produce different results depending on how the N-M alignment groups are broken up. It is difficult to know exactly how to break up an alignment group in Method B without knowing the annotator's intentions, so heuristic methods were used. For these results, the alignment groups were broken down such that simple sentences that were aligned to all complex sentences or vice versa were first grouped together. Then, the rest of the alignments were split into 1-N or N-1 groups.

## E Automatic Alignment Experiments

While automatic sentence alignment is standard in text simplification (Zhang and Lapata, 2017; Xu et al., 2015), we are not aware of evaluation work for sentence alignment algorithms on medical texts. The construction of MC-NOISY allows us to create a tiered version of MULTICOCHRANE that is large enough for training state-of-the-art neural text simplification models.

We evaluate a series of automatic sentence alignment methods to determine which approach is the most compatible with the medical data at hand to derive the English portion of MC-NOISY.

- **Jaccard Similarity**. The Jaccard Similarity score of every possible sentence pair was calculated. We consider pairs that score over a

threshold of 0.3 (selected on the validation set) to be "aligned"; this is the alignment method used in Xu et al. (2015).

- **TF-IDF Similarity**. We generated TF-IDF vectors for every possible pair. Pairs were scored based on the cosine similarity of their embeddings. We aligned each sentence in the PLS to the most similar sentence in the corresponding complex version. This is the alignment method used in Paetzold et al. (2017).

- **Sentence-BERT Similarity**. Similar to TF-IDF, but here we use Sentence-BERT embeddings (Reimers and Gurevych, 2019) instead.[6]

- **BERTScore**. The BERTScore (Zhang et al., 2020) between the complex and simplified sentences was calculated for every possible pair, and like for TF-IDF and Sentence-BERT, we considered the highest-scoring pair for each sentence in the PLS to be aligned.

- **Neural CRF Alignment**. We also evaluate a BERT-based CRF alignment model (Jiang et al., 2020). We test both the original model trained on the Newsela dataset, and a model fine-tuned on the training set of MC-CLEAN. The hyperparameters used for fine-tuning are the same those used for the original model.

For experiments, we used 395 alignments from 22 MC-CLEAN English articles as the test set, and 84 aligned sentence pairs from 5 articles as the validation set. The rest of the 1149 alignments from 74 articles were used for fine-tuning the neural CRF alignment model (Jiang et al., 2020). Table 9 reports the precision, recall, and F1 scores of these methods against reference annotated alignments in MC-CLEAN. We observe that the CRF alignment model trained on the Cochrane dataset achieved the best F1 measure; this is unsurprising, as it is the state-of-the-art supervised method for sentence alignment.

## F Human Evaluation Correlation

Table 10 displays the Pearson correlation between automatic metrics and the human evaluation scores. Overall the correlation between them is mostly

---

[6]For both TF-IDF and Sentence-BERT, we tested a version where a similarity threshold is selected; this led to much worse F1 scores, likely due to similarity being used as an absolute measure as opposed to a relative one.

| | Auto Metric | Fact. | Fluency | Simpl. |
|---|---|---|---|---|
| English | BLEU | -0.124 | -0.032 | 0.037 |
| | BERTScore | -0.174 | -0.174 | -0.030 |
| | SARI | -0.078 | 0.033 | 0.014 |
| | LCS | 0.319 | 0.144 | -0.254 |
| Spanish | BLEU | -0.190 | -0.100 | 0.017 |
| | BERTScore | -0.105 | -0.083 | 0.010 |
| | SARI | -0.037 | 0.091 | -0.367 |
| | LCS | 0.196 | 0.095 | -0.395 |
| French | BLEU | -0.197 | -0.108 | 0.042 |
| | BERTScore | -0.219 | -0.086 | 0.047 |
| | SARI | 0.029 | 0.035 | -0.186 |
| | LCS | 0.392 | 0.188 | -0.253 |
| Farsi | BLEU | -0.120 | -0.059 | 0.055 |
| | BERTScore | -0.281 | -0.074 | 0.227 |
| | SARI | -0.243 | 0.031 | 0.286 |
| | LCS | 0.319 | 0.257 | -0.347 |

Table 10: Pearson correlation between automatic metrics and human evaluation scores. Human evaluation metrics: closer to 0 = better; BLEU/BERTScore/SARI: higher = better; LCS: higher = more abstractive.

| Language | BLEU | BS | SARI | LCS |
|---|---|---|---|---|
| English | 0.25 | 0.8442 | 43.677 | 0.695 |
| Spanish | 1.40 | 0.7055 | 41.009 | 0.651 |
| French | 0.90 | 0.6806 | 41.987 | 0.695 |
| Farsi | 1.00 | 0.6862 | 44.204 | 0.698 |

Table 11: Automatic metrics for mT5 fine-tuned only on MC-Clean data.

weak. BLEU and BERTScore, not surprisingly, does have a weak inverse correlation with factuality and fluency measures. The metric with the strongest correlation to human scores is LCS, showing a moderate positive correlation with factuality (more abstractive sentences tend to contain more factual errors), as well as a moderate inverse correlation with the simplicity score (more abstractive sentences tend to be simpler). The other simplification metric, SARI, had varied results depending on the language.

## G    Fine-tuning with MC-CLEAN Only

We attempted fine-tuning mT5 with just data from MC-CLEAN. The same training methodology was used, with the only difference being the number of epochs being increased to 5. Overall, due to the significantly lower amount of alignments in MC-CLEAN as compared to MC-NOISY, the generated output featured many extreme hallucinations. The automatic metrics for these generations is shown in Table 11. The low BLEU score and BERTScore

| | Dataset | en→en | en→es | en→fr | en→fa |
|---|---|---|---|---|---|
| Complex | MC-NOISY | 39.7 | 39.5 | 39.4 | 41.1 |
| | + filtering | 44.8 | 44.5 | 44.4 | 46.6 |
| | MC-CLEAN | 41.6 | 41.6 | 41.7 | 43.2 |
| Simple | MC-NOISY | 36.0 | 48.0 | 54.1 | 52.7 |
| | + filtering | 32.3 | 43.6 | 49.3 | 48.0 |
| | MC-CLEAN | 36.6 | 47.2 | 55.0 | 49.5 |

Table 12: Statistics for average number of tokens per sentence for complex and simple sentences in MC-NOISY and MC-CLEAN.

are the results of this hallucinating behavior.

The following is an example of an extreme hallucination found among the generated outputs for the English version.

> *Input complex sentence*: Randomised controlled trials of tai chi lasting at least three months involving healthy adults or adults at high risk of CVD.
> *Output*: Another study investigated the effect of physical exercise as a treatment for people with carpal tunnel syndrome.

While the output does mention a "study" and the use of "physical exercise," which is somewhat related to a randomized controlled trial about tai chi, it remains wildly off-topic compared to the original sentence.

## H    Additional Statistics

Table 12 shows the statistics for the average number of tokens per sentence in MULTICOCHRANE.

# I Notable Examples

## I.1 Good Simplifications

Below are some examples of good simplifications made by the systems. GPT3 systems in particular were good at explaining medical terms and information present in parentheses. Meanwhile, other systems like Flan-T5-ft (r=0.5) and Flan-T5 (0-shot) had fluency issues and hallucinations even in the best simplifications and had more difficulty with longer input sentences. mT5 outputs had issues with extractiveness and retained complex medical vocabulary.

| Language | System | Input | Output | Gloss |
|---|---|---|---|---|
| English | GPT3 (0-shot) | We also found improved success rates within two attempts (RR 1.78, 95% CI 1.25 to 2.51, 134 catheters, two RCTs, moderate-quality evidence) with ultrasound guidance compared with other types of guidance. | This sentence is talking about a study that was done to compare two different types of guidance when using a medical tool called a catheter. The study found that if ultrasound guidance was used, it was more successful than other types of guidance. Out of 134 catheters that were used, the success rate was 1.78 times higher when ultrasound guidance was used. This result was reliable because two different tests were done, and the evidence was of moderate quality. | - |
| Spanish | GPT3 (0-shot) | Four RCTs compared ultrasound with palpation, and one compared ultrasound with Doppler auditory assistance. | Cuatro estudios de investigación evaluaron la eficacia de usar ultrasonido para detectar problemas médicos, comparándolo con usar la palpación (tocar con las manos) y uno de los estudios comparó el uso del ultrasonido con ayuda de un aparato llamado Doppler para escuchar los sonidos del cuerpo. | Four investigative studies evaluated the efficacy of using ultrasound to detect medical problems, comparing it to using the palpation (touching with the hands) and one of the studies compared the use of the ultrasound with the help of an apparatus called Doppler to listen to the sounds of the body. |
| English | mT5 (r=0.5) | There were no serious adverse events related to either nitrous oxide or carbon dioxide pneumoperitoneum (three studies; 196 participants; very low quality of evidence). | There were no serious side effects related to either nitrous oxide or carbon dioxide pneumoperitoneum. | - |
| French | mT5 (r=0.5) | Thiospot was more effective than placebo (SMD -2.61, 95% CI -3.76 to -1.47). | La thiospots était plus efficace que le placebo. | Thiospot was more effective than the placebo. |
| Spanish | GPT3-simp-trans | There were no serious adverse events related to either nitrous oxide or carbon dioxide pneumoperitoneum (three studies; 196 participants; very low quality of evidence). | Los tres estudios realizados con 196 personas mostraron que no hubo efectos negativos al respirar gas nitrógeno o dióxido de carbono durante un procedimiento médico. | The three studies carried out with 196 people showed that there were no negative effects in breathing nitrogen gas or carbon dioxide during a medical procedure. |

| Language | System | Input | Output | Gloss |
|---|---|---|---|---|
| French | GPT3-simp-trans | To assess interventions used in the management of all types of melasma: epidermal, dermal, and mixed. | Pour déterminer quels traitements fonctionnent le mieux pour les différents types de mélasma (une affection cutanée), comme le type qui affecte la couche supérieure de la peau, le type qui affecte la couche sous la couche supérieure et le type qui affecte les deux couches. | In order to determine which treatments work the best for the different types of melasma (a skin affliction), like the type which affects the top layer of the skin, the type which affects the layer under the top layer and the type which affects the two layers. |
| Spanish | mT5 (r=0) | For smokeless tobacco use, no difference was found between the skills-community arm and the control group at 42 weeks (RR 0.93, 95% CI 0.67 to 1.30), though a significant difference was observed between the skills-only arm and the control group (RR 0.57, 95% CI 0.39 to 0.85). | En cuanto a los fumadores sin cigarrillos, se identificaron pocos casos y no hubo diferencias entre el grupo de ejercicios y el grupo control a las 42 semanas, pero se observaron efectos secundarios entre el grupo de ejercicios y el grupo de control. | As for smokers without cigarettes, few cases were identified and there were no differences between the exercise group and the control group at 42 weeks, but side effects were observed between the exercise group and the control group. |
| French | mT5 (r=0) | We searched the Cochrane Stroke Group Trials Register (16 January 2018), the Cochrane Central Register of Controlled Trials (CENTRAL; 2017, Issue 12) in the Cochrane Library (January 2018), MEDLINE Ovid (1946 to January 2018), Embase Ovid (1974 to January 2018), Web of Science Core Collection (1950 to January 2018) and three Chinese databases (January 2018). | Nous avons recherché des études complémentaires publiées jusqu'au 16 janvier 2018. | We researched additional published studies until the 16 of January 2018. |
| Spanish | Flan-T5-ft (r=0.5) | We included seven trials, involving 735 participants, in this review. | Se incluyeron siete ensayos, que incluyeron 735 participantes. | Seven trials, involving 735 participants, were included. |
| French | Flan-T5-ft (r=0.5) | The quality of the evidence was rated as high. | La qualité des preuves était élevée. | The quality of the evidence was high. |
| Spanish | Flan-T5 (0-shot) | We searched until January 2016. | También buscamos hasta enero 2016. | We also searched until January 2016. |
| English | Flan-T5 (0-shot) | We searched the Cochrane Kidney and Transplant Specialised Register up to 17 August 2017 through contact with the Information Specialist using search terms relevant to this review. | Cochrane Kidney and Transplant Specialised Register was searched for up to 17 August 2017. | |

Table 13

## I.2 Fluency

Fluency errors across systems include repetition and grammatical errors, as well as the use of foreign words in the Flan outputs.

**Repetition.** This category includes examples of fluency errors involving the repetition of a single phrase or word that confuses the meaning of the output and, in some cases, makes it completely meaningless. This error was present in all systems except for the GPT3 (0-shot) and GPT3-simp-trans systems.

| Language | System | Input | Output | Gloss |
|---|---|---|---|---|
| English | mT5 (r=0) | In people with symptomatic intracerebral artery stenosis, recurrent stroke was significantly reduced by RIC (risk ratio (RR) 0.32, 95% confidence interval (CI) 0.12 to 0.83; 2 trials, 182 participants, low-quality evidence). | Compared to other antipsychotic drugs, recurrent stroke was significantly reduced by recurrent stroke. | - |
| English | mT5 (r=0.5) | Similarly, our results suggest, but do not confirm, the possibility of a positive influence of expertise in the use of ultrasound on the first attempt success rate. | A significantly significantly significant difference in the use of ultrasound on the first attempt success rate. | - |
| Spanish | Flan-T5-ft (r=0.5) | We located 10 RCTs involving 2961 participating surgeons performing an operation in which the use of blunt needles was compared to the use of sharp needles. | Se identificaron 10 ECA con 2961 médicos que realizaron una operación que fue realizada con agujas y cuando se comparó agujas tópicas con agujas tópicas y con agujas agujas. | We identified 10 RCTs with 2961 doctors who performed an operation that was performed with needles and when topical needles were compared with topical needles and with needles needles. |
| Spanish | mT5 (r=0) | Results of the ongoing trials will add to the evidence base but additional longer-term, high-quality trials are needed. | Los resultados de los ensayos en curso adicionales a corto plazo, adicionales, de alta calidad, se necesitan más estudios de alta calidad. | The results of the ongoing, additional short-term trials, additional [trials], of high quality, more high-quality studies are needed. |
| French | mT5 (r=0) | Smoking prevalence in Indigenous youth is twice that of the non-Indigenous population, with tobacco experimentation commencing at an early age. | La prévalence du tabagisme chez les jeunes jeunes jeunes développent des problèmes d'éstomac à l'avenir et présentent une amélioration des symptômes. | The prevalence of smoking in the youth youth youth developed stomach problems in the future and presents an improvement of the symptoms. |
| French | mT5 (r=0.5) | The rates were 24.5 for the sponge and 10.9 for the diaphragm in the UK trial. | Les taux étaient étaient 25 pour le tube de sponge et 10,9 pour le diaphragme au Royaume-Uni. | The rate was was 25 for the sponge tube and 10.9 for the diaphragm in the UK. |

Table 14

**Grammar.** This category includes examples of grammatical issues in the output sentences, such as the overuse of prepositions, spelling errors, and disagreement between articles and nouns. Multiple prepositions appeared sequentially in a way that was incoherent, and there were instances where articles did not agree with nouns in gender or number.

| Language | System | Input | Output | Gloss |
|---|---|---|---|---|
| French | mT5 (r=0) | There appears to be no significant differences in the improvement of technical skills between different methods of box model training. | En ce qui concerne les modifications de la technique manuelle des méthodes de sélection de modèles bovés, la majorité des participants dans leur étude n'ont montré pas de différence significative entre les méthodes d'entrainement au modèle d'étudiant de modèle d'évaluation en termes d'amélioration des compétences informatiques pour différentes méthodes d'entraînement au modèle bouchon. | Concerning the modifications of the manual technique of the selection methods of bové(?) models, the majority of participants in their study didn't show a significant difference between the training methods in the student model of the evaluation model in terms of improvement of informatic skills for the different training methods in the blockage model. |
| Spanish | Flan-T5-ft (r=0.5) | carbon dioxide, helium, argon, nitrogen, nitrous oxide, and room air) used for establishing pneumoperitoneum in participants undergoing laparoscopic general abdominal or gynaecological pelvic surgery. | El uso de los dióxidos carbonosas, el alto carbono, la argón, la náusea y el fármaco nitroso (es decir, los flujos náuseos y el fármaco nitroso). | The use of the carbon dioxides, the high carbon, the argon, the nausea and the nitrous drug (which is to say, the nausea flows and the nitrous drug). |
| Spanish | Flan-T5-ft (r=0.5) | Indigenous youth in particular commence smoking at an early age, and a disproportionate burden of substance-related morbidity and mortality exists as a result. | Es probable que la morbilidad de los nios de ingesta a un término temprano o a un pas a tener un empeoramiento de salud y muerte debido a una pérdida del tratamiento. | It is likely that the morbidity of the chiren of intake to an early term or a ste to have a worsening of health and death due to a loss of the treatment. |

Table 15

**Foreign words (Flan-T5).** This category includes examples of fluency errors involving the use of foreign words in the output sentences. For example, Flan-T5 (0-shot) produced a French output that started with the English phrase 'in four studies', and Flan-T5-ft (r=0.5) produced an English output that started with the Spanish word 'otros'. This type of error was mostly found in the Flan-T5-ft (r=0.5) and Flan-T5 (0-shot) systems.

| Language | System | Input | Output | Gloss |
|---|---|---|---|---|
| French | Flan-T5-ft (r=0.5) | It usually involves restricting blood flow to limbs and then releasing the ischaemic blood to promote a neuroprotective effect. | L'exercice consiste généralement à la restriction du flux sanguin de la membres des limbs et ensuite à libérer les bloods ischémies pour promouvoir un effet neuroprotecteur. | The exercise generally involves the restriction of the blood flow to the members of the limbs and then to free the ischemic blood to promote a neuroprotective effect. |
| French | Flan-T5 (0-shot) | In four studies, the use of blunt needles reduced the number of self-reported needle stick injuries with a RR of 0.31 (95% CI 0.14 to 0.68). | In four studies, l'utilisation des nejéros strru00f7u a réduit l'injurage autoremprésenté aux nejéros d'une rRR de 0,31, en moyenne de 0,14 à 0,68. | In four studies, the use of nejéros strru00f7u (?) to reduce the autoremprésenté (?) injury to the nejéros (?) of a risk ratio of .31, on average from .14 to .68. |
| English | Flan-T5-ft (r=0.5) | Therefore, other gases have been introduced as alternatives to carbon dioxide for establishing pneumoperitoneum. | Otros such as carbon dioxide are used as alternative gases for establishing pneumoperitoneum. | - |

| Language | System | Input | Output | Gloss |
|---|---|---|---|---|
| Spanish | Flan-T5 (0-shot) | Remote ischaemic conditioning (RIC) has been developed as a neuroprotective strategy to prevent and treat ischaemic stroke. | El afecto izquierdo de condicionamiento remoto (RIC) ha sido desarrollado como una estrategia neuro-promotiva para prevenir y tratar strokes izquierdo. | The remote conditioning left effect (RIC) has been developed as a neutropromotive to help prevent and treat left strokes. |

Table 16

## I.3 Factuality

Factuality errors displayed by the systems could be broadly categorized into 3 types - minor hallucinations, extreme hallucinations, and contradictions.

**Minor hallucinations.** This category includes examples where the output contained minor details that were not present in the input (hallucination). The systems would hallucinate details like numbers, dates, medical terms, incorrect expansions of acronyms, etc. GPT3 systems would also misinterpret the context of words present in the input. In one of the examples, the word 'rates' in the input is referring to 'pregnancy rates' but GPT3 misinterpreted the word to mean 'cost'.

| Language | System | Input | Output | Gloss |
|---|---|---|---|---|
| French | mT5 (r=0) | To assess interventions used in the management of all types of melasma: epidermal, dermal, and mixed. | Cette revue a examiné différents type de mélanome, de la peau, d'œil et des organes, et s'est améliorée dans des années futures. | This review has examined different types of melanoma, of the skin, of the eye and of the organs, and improved in the future years. |
| Spanish | GPT3 (0-shot) | It usually involves restricting blood flow to limbs and then releasing the ischaemic blood to promote a neuroprotective effect. | Esta oración está hablando de un procedimiento médico llamado Prea-condicionamiento Isquémico (IPC). El procedimiento consiste en cortar temporalmente el flujo de sangre a ciertas partes del cuerpo y luego devolver la sangre al cuerpo. El objetivo de este procedimiento es proteger el cerebro y el sistema nervioso del daño. | This sentence is talking about a medical procedure called Ischemic Preconditioning (IPC). The procedure consists of temporarily cutting off the flow of blood to certain parts of the body and then returning the blood to the body. The objective of this procedure is to protect the brain and the nervous system from damage. |
| Spanish | mT5 (r=0.5) | We included five RCTs reporting 444 arterial cannulations in paediatric participants. | Se incluyeron cinco ensayos controlados aleatorios que informaron 556 cannulaciones arteriales en los participantes niños. | Five randomized controlled trials that reported 556 arterial cannulations in pediatric participants were included. |
| French | mT5 (r=0) | The rates were 24.5 for the sponge and 10.9 for the diaphragm in the UK trial. | Les taux étaient de 18,3 pour le diaphragme et de 10,9 pour le diaphragme avec un diaphragme UK dans l'essai. | The rates were 18.3 for the diaphragm and 10.9 for the diaphragm with a UK diaphragm in the trial. |
| English | mT5 (r=0) | We included three trials (involving 371 participants) in the analysis of the effects of RIC on ischaemic stroke prevention. | We included three trials with 373 participants and provided one trial with seven trials involving 373 participants. | - |

| Language | System | Input | Output | Gloss |
|---|---|---|---|---|
| French | mT5 (r=0) | The adverse events most commonly reported were mild and transient such as skin irritation, itching, burning, and stinging. | Les effets secondaires les plus fréquemment signalés étaient les plus graves et transitoires : la irritation cutanée et le fuite d'eau, les brûlures, les vertiges et les vertiges. | The secondary effects most frequently reported were the most serious and transitory: skin irritation and the water leak, burns, dizziness and dizziness. |
| English | mT5 (r=0) | We searched the Cochrane Kidney and Transplant Specialised Register up to 17 August 2017 through contact with the Information Specialist using search terms relevant to this review. | We searched scientific databases for studies on 27 March 2017. | - |
| French | mT5 (r=0.5) | We included seven trials, involving 735 participants, in this review. | Nous n'avons trouvé aucun essai, avec un total de 535 participants. | We haven't found a trial, with a total of 535 participants. |
| English | Flan-T5-ft (r=0.5) | Our results suggest, but do not confirm, that a possible advantage of ultrasound guidance for the first attempt success rate over other techniques is more pronounced in infants and small children than in older children. | Furthermore, ultrasound guidance is more accurate and safer than alternative techniques for treating first tentative baby ear infection. | - |
| English | GPT3 (0-shot) | The rates were 24.5 for the sponge and 10.9 for the diaphragm in the UK trial. | In the UK trial, it cost 24.5 pounds to buy a sponge and 10.9 pounds to buy a diaphragm. | - |

Table 17

**Extreme hallucinations.** This category includes examples of hallucinations that were extremely irrelevant. Information hallucinated in the output sentences has no connection to the information present in the input. This degree of hallucinations was much more prevalent and extreme in the Flan systems as compared to other systems.

| Language | System | Input | Output | Gloss |
|---|---|---|---|---|
| English | Flan-T5 (0-shot) | Randomized controlled trials based on uniform and standard methods are needed. | The aim of the experiments is to give patients a chance to perform the experiment they want to do. | - |
| French | Flan-T5-ft (r=0.5) | In acute ischaemic stroke, for people receiving intravenous thrombolysis, the rate of death or dependency was significantly increased by RIC treatment compared with non-RIC treatment (RR 2.34; 95% 1.19 to 4.61; 1 trial, 285 participants, low-quality evidence). | Pour les personnes victimes d'un AVC aigu, les femmes admissées de la greffe avait une probabilité accrue d'être victimes ou de la maladie grâce à un traitement remmanant par injection d'une coagulation récurrente. | For the victims of acute stroke, the women administered (?) the transplant had an increased probability of being victims or of the disease thanks to a remmanant (?) treatment by an injection of a recurrent coagulation. |

| Language | System | Input | Output | Gloss |
|---|---|---|---|---|
| Spanish | mT5 (r=0.5) | In people with carotid stenosis undergoing carotid stenting, there was no significant difference in the incidence of ischaemic stroke between participants treated with RIC and non-RIC (RR 0.22, 95% CI 0.01 to 4.03; 1 trial, 189 participants, low-quality evidence); however the stroke severity (assessed by infarct volume) was significantly lower in participants treated with RIC (mean difference (MD) -0.17 mL, 95% CI -0.23 to -0.11; 1 trial, 189 participants, low-quality evidence). | En los pacientes con enfermedad isquémica cuando se acortan los brazos en los pacientes tratados con ENIC y ningún riesgo de caídas. | In the patients with ischemic disease, when arms are shorted in the patients treated with ENIC and no risk of falls. |

Table 18

**Contradictions.** This category includes examples where the output directly contradicts ideas stated in the input. This error was infrequent in the GPT3 systems but was equally prevalent in other systems.

| Language | System | Input | Output | Gloss |
|---|---|---|---|---|
| French | Flan-T5 (0-shot) | There was considerable heterogeneity between the studies included in this review and studies were small and at some risk of bias | Il y a lieu d'échantillonner des études incluant dans cette étude et les études étaient grandes et en hausse une hausse de la biase. | There is a reason to sample the studies included in this study and the studies were big and increasing an increase of bias. |
| English | mT5 (r=0) | There is qualitative evidence that oral morphine has much the same efficacy as other available opioids. | It is not clear if oral morphine is as effective as other available opioids. | - |
| French | mT5 (r=0.5) | In people with acute ischaemic stroke, there was no significant difference between RIC and non-RIC for reducing stroke severity as assessed by the National Institutes of Health Stroke Scale score and the final infarct volume (standardised mean difference (SMD) -0.24 mL, 95% CI -1.02 to 0.54; 2 trials, 175 participants, very low quality evidence). | La SEP réduisait la gravité du AVC ischémique, mais les essais confondus avaient des preuves de très faible qualité. | The SEP reduced the severity of ischemic stroke, but the included trials have evidence of very weak quality. |
| Spanish | mT5 (r=0.5) | Our results suggest, but do not confirm, that a possible advantage of ultrasound guidance for the first attempt success rate over other techniques is more pronounced in infants and small children than in older children. | Los resultados indican que la guía ecográfica redujo la tasa de éxito del ensayo más amplia en los lactantes y los niños pequeños en comparación con otras técnicas. | The results indicate that the ultrasound guidance reduced the success rate of the broader trial in the infants and the young children in comparison with other techniques. |

| Language | System | Input | Output | Gloss |
|---|---|---|---|---|
| English | mT5 (r=0.5) | There is qualitative evidence that oral morphine has much the same efficacy as other available opioids. | The evidence for this topic is current to December 2018 that oral morphine has a serious toxicity as a different opioid. | - |

<p align="center">Table 19</p>

## I.4 Simplification

Simplification errors included instances of the system being overly extractive, oversimplification of terms and ideas, and excessive deletion of entire parts of the original text.

**Extractive.** GPT3 (0-shot) was highly extractive and failed to explain complex medical language in everyday terms that a general audience could understand. The system also left in statistics from the input sentence that are not interpretable to the general public. mT5 (r=0) exhibited similar patterns with regards to extractiveness.

| Language | System | Input | Output | Gloss |
|---|---|---|---|---|
| Spanish | GPT3 (0-shot) | Interventions included: thrombolytic therapy versus placebo (1 study); low versus high dose thrombolytic therapy (1); alteplase versus urokinase (1); short versus long thrombolytic dwell (1); thrombolytic therapy versus percutaneous fibrin sheath stripping (1); fibrin sheath stripping versus over-the-wire catheter exchange (1); and over-the-wire catheter exchange versus exchange with and without angioplasty sheath disruption (1). | Esta frase describe un estudio que se hizo para comparar diferentes tratamientos para ayudar a mejorar la circulación sanguínea. Estos tratamientos incluyeron: terapia con trombolíticos en comparación con un placebo (1 estudio); dosis baja en comparación con dosis alta de trombolíticos (1); alteplasa en comparación con uroquinasa (1); tiempo corto de trombolíticos en comparación con tiempo largo (1); terapia con trombolíticos en comparación con despeje de fibrina con una aguja (1); despeje de fibrina en comparación con cambio de catéter con alambre (1); y cambio de catéter con alambre en comparación con cambio sin despeje de fibrina con angioplastia (1). | This sentence describes a study that was conducted to compare different treatments to help improve the blood circulation. These treatments included: therapy with thrombolytics in comparison with a placebo (1 study); low dose in comparison with high dose of thrombolytics (1); alteplase compared to urokinase (1); short time of thrombolytics compared with long time (1); therapy with thrombolytics compared to clearance of fibrin with a needle (1); clearance of fibrin in comparison with wire catheter change (1); and wire catheter change in comparison with change without clearance of fibrin with angioplasty (1). |
| Spanish | mT5 (r=0) | On current data there is no evidence to support physical intervention over the use of pharmaceutical agents in the acute setting. | Esta evidencia no apoya la intervención física sobre la administración de agentes farmacológicos en la práctica aguda. | This evidence does not support the physical intervention over the administration of pharmaceutical agents in the acute practice. |

| Language | System | Input | Output | Gloss |
|---|---|---|---|---|
| French | GPT3 (0-shot) | In acute ischaemic stroke, for people receiving intravenous thrombolysis, the rate of death or dependency was significantly increased by RIC treatment compared with non-RIC treatment (RR 2.34; 95% 1.19 to 4.61; 1 trial, 285 participants, low-quality evidence). | En cas de crise d'ischémie aiguë, pour les personnes qui ont reçu un traitement par thrombolyse par voie intraveineuse, le taux de décès ou de dépendance a été significativement augmenté par le traitement RIC par rapport au traitement non RIC (RR 2,34; 95% 1,19 à 4,61; 1 essai, 285 participants, preuves de faible qualité). | In the case of an acute ischemic crisis, for the people who have received a treatment by intravenous thrombolysis, the rate of death or dependence was significantly increased by the RIC treatment in comparison to the non-RIC treatment (RR 2.34; 95% 1.19 to 4.61; 1 trial, 285 participants, evidence of weak quality). |

Table 20

**Oversimplification**  Flan-T5-ft (r=0.5) had issues with excessive deletions, resulting in fragmented sentences that failed to communicate the meaning of the input sentence. Excessive deletion in mT5 (r=0) and mT5 (r=0.5) manifested as a loss of information that was necessary to preserve the meaning of the input sentence. The resulting outputs were missing critical information, causing them to be generic and vague, such as in the Spanish mT5 (r=0.5) example that discusses the use of RIC for ischemic stroke prevention.

| Language | System | Input | Output | Gloss |
|---|---|---|---|---|
| French | Flan-T5-ft (r=0.5) | We rated the quality of evidence for all outcomes as moderate owing to imprecision due to wide confidence intervals, modest sample sizes and limited numbers of events. | La qualité des preuves | The quality of the evidence |
| French | mT5 (r=0) | Four RCTs compared ultrasound with palpation, and one compared ultrasound with Doppler auditory assistance. | Cinq ECR ont comparé l'échographie avec une palpation. | Five RCTs have compared ultrasounds with palpation. |
| French | mT5 (r=0.5) | We included three trials (involving 371 participants) in the analysis of the effects of RIC on ischaemic stroke prevention. | Les auteurs de la revue ont inclus un total de 371 participants. | The authors of the review included a total of 371 participants. |
| Spanish | mT5 (r=0.5) | The quality of the evidence was rated as high. | Calidad de la evidencia | Quality of the evidence |
| Spanish | mT5 (r=0.5) | We included three trials (involving 371 participants) in the analysis of the effects of RIC on ischaemic stroke prevention. | Se incluyeron tres ensayos que incluyeron a 372 participantes. | Three trials with 372 participants were included. |
| English | mT5 (r=0) | There were no serious adverse events related to either nitrous oxide or carbon dioxide pneumoperitoneum (three studies; 196 participants; very low quality of evidence). | There were no serious adverse events. | - |

| Language | System | Input | Output | Gloss |
|---|---|---|---|---|
| English | mT5 (r=0.5) | We reran the searches for all databases in March 2016, and when we update the review, we will deal with the two studies of interest found through this search that are awaiting classification. | The search for most scientific databases in March 2016. | - |

Table 21

## I.5 Other Errors

**Repetition of output (GPT3 (0-shot)).** The GPT3 (0-shot) system had a unique error that exclusively appeared in the French outputs where a separate paraphrase was generated in addition to the main output sentence. The system sometimes gave a generated output, then repeated the same information in different words using the phrase 'in other words' or simply generating two different versions of a simplified output.

| Language | System | Input | Output | Gloss |
|---|---|---|---|---|
| French | GPT3 (0-shot) | All but one trial were at high risk of bias. | Tous les essais sauf un étaient à haut risque de biais. En d'autres mots, la plupart des essais présentaient un risque élevé d'être biaisés. | All the trials except one were at high risk of bias. In other words, the majority of the trials presented an elevated risk of being biased. |
| French | GPT3 (0-shot) | We rated the quality of evidence for all outcomes as moderate owing to imprecision due to wide confidence intervals, modest sample sizes and limited numbers of events. | Nous avons estimé que la qualité de la preuve pour tous les résultats était modérée en raison d'une imprécision due à des intervalles de confiance larges, des tailles d'échantillon modestes et à un nombre limité d'événements. Nous avons estimé que la preuve était pas très précise parce que les intervalles de confiance étaient grands, les échantillons pas très grands et il y avait peu d'événements. | We estimated that the quality of the evidence for all the results was moderate on account of an imprecision due to large confidence intervals, the size of the modest samples and to a limited number of events. We estimated that the evidence wasn't very precise because the confidence intervals were big, the not very big samples and there were few events. |

Table 22

**Acronyms.** Across all systems, a frequent issue that arose when simplifying medical text was acronyms. These issues can be further broken down into failure to translate into the target output language, hallucinated acronym generations, and factually incorrect generations.

**Acronym translation.** Systems failed to translate the input sentence's acronym from English into Spanish or French. GPT3 (0-shot) would explicitly state that the acronym was in English, rather than translating, in some cases.

| Language | System | Input | Output | Gloss |
|---|---|---|---|---|
| Spanish | GPT3-simp-trans | To assess the benefits and harms of RIC for preventing ischaemic stroke and for treating people with ischaemic stroke and those at risk for ischaemic stroke. | Esta oración nos pide que averigüemos si cierto tipo de tratamiento médico (RIC, por sus siglas en inglés) es útil para prevenir y tratar los accidentes cerebrovasculares. Queremos saber si los beneficios de usar RIC son mayores que los riesgos. | This sentence is asking us to find out if a certain medical treatment (RIC, by its acronym in English) is useful to prevent and treat strokes. We want to know if the benefits of using RIC are greater than the risks. |
| French | GPT3 (0-shot) | We included randomised controlled trials (RCTs) comparing RIC with sham RIC or medical management in people with ischaemic stroke or at risk of ischaemic stroke. | Nous avons inclus des essais contrôlés randomisés (RCT) pour comparer le RIC avec un faux RIC ou une gestion médicale chez les personnes ayant subi un accident vasculaire cérébral ou à risque d'accident vasculaire cérébral. | We have included randomly controlled trials (RCT) to compare the RIC with a fake RIC or medical management in people that suffered a stroke or at risk of a stroke. |
| French | Flan-T5 (0-shot) | To assess the benefits and harms of RIC for preventing ischaemic stroke and for treating people with ischaemic stroke and those at risk for ischaemic stroke. | évaluer les avantages et les éléments de la RIC en vue de supprimer le stroke insécutif et en vue de réduire les risques liées à l'insécutif. | To evaluate the advantages and the elements of RIC in order to remove the insécutif (?) stroke and in order to reduce the risks associated with the insécutif (?). |

Table 23

**Acronym hallucination.** Flan-T5-ft (r=0.5) hallucinated acronyms in the output sentence that do not exist in the target language. This occurred across all languages and in both Flan and mT5 systems.

| Language | System | Input | Output | Gloss |
|---|---|---|---|---|
| English | Flan-T5-ft (r=0.5) | In acute ischaemic stroke, for people receiving intravenous thrombolysis, the rate of death or dependency was significantly increased by RIC treatment compared with non-RIC treatment (RR 2.34; 95% 1.19 to 4.61; 1 trial, 285 participants, low-quality evidence). | For acute ischaemic stroke, for people receiving intravenous thrombolysis, the rate of death or dependency was significant increased by RAC treatment compared with non-RAC treatment (low-quality evidence). | |
| French | Flan-T5-ft (r=0.5) | Preclinical studies have suggested that RIC may have beneficial effects in ischaemic stroke patients and those at risk of ischaemic stroke. | Les résultats de cette revue ont suggéré que les BRA pourrait avoir des effets bénéfiques dans les patients souffrant d'une PIO et chez les personnes ayant un accident cérébral ischémique. | The results of this review suggested that the BRA could have beneficial effects in patients suffering a PIO and in people who had an ischemic stroke. |

| Language | System | Input | Output | Gloss |
|----------|--------|-------|--------|-------|
| French | mT5 (r=0.5) | In people with acute ischaemic stroke, there was no significant difference between RIC and non-RIC for reducing stroke severity as assessed by the National Institutes of Health Stroke Scale score and the final infarct volume (standardised mean difference (SMD) -0.24 mL, 95% CI -1.02 to 0.54; 2 trials, 175 participants, very low quality evidence). | La SEP réduisait la gravité du AVC ischémique, mais les essais confondus avaient des preuves de très faible qualité. | The SEP reduced the severity of ischemic stroke, but the included trials have evidence of very weak quality. |

Table 24

**Acronym factuality.** Flan-T5 (0-shot) and mT5 (r=0) produced factuality issues in the output sentences. In the Spanish output example, Flan generated the acronym for AIDS (SIDA), though the input was referring to randomized controlled trials (RCTs). In the French example, mT5 (r=0) generated the acronym for a stroke (AVC), but the input sentence refers to cardiovascular disease (CVD).

| Language | System | Input | Output | Gloss |
|----------|--------|-------|--------|-------|
| Spanish | Flan-T5 (0-shot) | Eight RCTs with 733 women in total that compared brief co-incubation and the standard insemination protocol were included. | Ocho estudios compararon la co-incubación breve con un protocolo de inserción estándar en pacientes con SIDA en total. | Eight studies compared the brief co-incubation with a protocol of standard insertion in patients with AIDS in total. |
| French | mT5 (r=0) | Stress and a sedentary lifestyle are major determinants of cardiovascular disease (CVD). | Le stress et le règlement inactif sont des causes importantes de maladies cardiovasculaires (AVC) les plus courants. | Stress and inactive regulation are the most common important causes of cardiovascular disease (AVC). |

Table 25

**Input hallucination (GPT3).** The systems GPT3-simp-trans and GPT3 (0-shot) show some minor hallucination by referencing the input sentence in the output. Some output generations would start with 'this phrase' or 'this sentence' in talking about the input. This error was not found in any of the other systems.

| Language | System | Input | Output | Gloss |
|---|---|---|---|---|
| English | GPT3 (0-shot) | There was no significant difference between RIC and non-RIC for improving the psychological impairment (SMD -0.37 points, 95% CI -1.15 to 0.41; 1 trial, 26 participants, very low quality evidence) and the cognitive impairment (SMD -0.26 points; 95% CI -0.72 to 0.21; 3 trials, 79 participants, low-quality evidence) in people with acute ischaemic stroke and cerebral small vessel disease. | This sentence is talking about two different groups of people who had strokes: those who received a type of treatment called RIC (RIC stands for Rehabilitation Intervention in Cerebrovascular Disease) and those who did not. It is talking about how the treatment affected their psychological and cognitive health. The results showed that there was not much difference between the two groups in terms of improving the psychological and cognitive health of the people with strokes. | - |
| French | GPT3 (0-shot) | In people with carotid stenosis undergoing carotid stenting, there was no significant difference in the incidence of ischaemic stroke between participants treated with RIC and non-RIC (RR 0.22, 95% CI 0.01 to 4.03; 1 trial, 189 participants, low-quality evidence); however the stroke severity (assessed by infarct volume) was significantly lower in participants treated with RIC (mean difference (MD) -0.17 mL, 95% CI -0.23 to -0.11; 1 trial, 189 participants, low-quality evidence). | Cette phrase parle d'une étude qui a été faite sur des personnes souffrant de sténose carotide et qui ont été traitées par stent carotidien. L'étude montre qu'il n'y a pas de différence significative entre les personnes traitées par RIC (Revascularisation Intracarotidienne) et celles qui ne l'ont pas été, au niveau du risque d'accident vasculaire cérébral. Cependant, les personnes traitées par RIC ont présenté une sévérité moindre des accidents vasculaires cérébraux (mesurée par le volume d'infarctus). | This phrase talks about a study that was done on people suffering from carotid stenosis and who were treated by carotid stenting. The study shows that there isn't a significant difference between people treated by RIC (Revascularization Intracarotid) and those who weren't, at the level of risk of stroke. However, the people treated by RIC presented a lessened severity of strokes (measured by the volume of infarction). |
| Spanish | GPT3-simp-trans | Three trials randomised participants to nitrous oxide pneumoperitoneum (100 participants) or carbon dioxide pneumoperitoneum (96 participants). | Esta oración describe un experimento científico que probó dos tipos diferentes de tratamientos médicos en dos grupos de personas. El primer grupo tenía 100 personas y se les dio un tratamiento con óxido nitroso. El segundo grupo tenía 96 personas y se les dio un tratamiento con dióxido de carbono. | This sentence describes a scientific experiment that tested two different types of medical treatment in two groups of people. The first group had 100 people and they were given a treatment with nitrous oxide. The second group had 96 people and they were given a treatment with carbon dioxide. |

Table 26

## J ANNO-VIEWER

ANNO-VIEWER is inspired by a similar annotation tool that was used in Jiang et al. (2020) for correcting crowdsourced alignment labels used for training and evaluating a neural CRF model for aligning sentences.

The primary function of this tool is to enable annotators to make alignments efficiently. ANNO-VIEWER also enables annotators to make factuality annotations for alignments. These annotations are largely based from the factuality annotations used in Devaraj et al. (2022), and also includes an additional field to annotate for elaborations in a similar manner.

This annotation tool also has the additional functionality of allowing annotators to use existing similarity

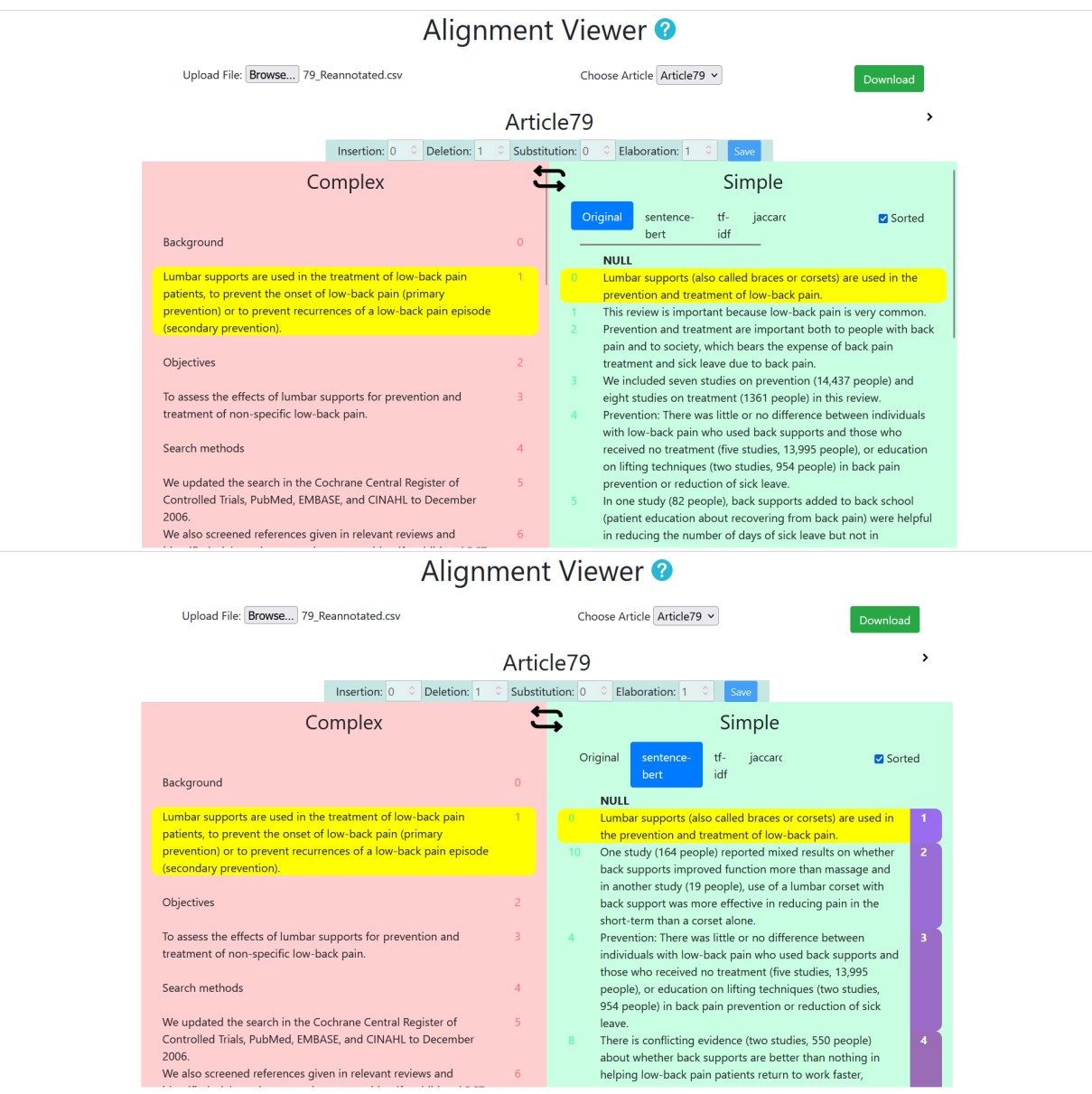

Figure 4: Screenshots of ANNO-VIEWER. The sentence sorting tool is featured in the bottom image.

measures to help find sentence alignments faster. When a sentence is selected for alignment, the tool can sort the other document by which sentences are most similar (determined by the automatic similarity measure) to the selected sentence.