# OpenReview forum: "Multilingual Simplification of Medical Texts"
_EMNLP/2023/Conference — EMNLP 2023 Main_

### Official Review · Reviewer_dMiC · 2023-08-03

**Soundness:** 4

**Ethical Concerns:**

Yes

**Excitement:**

4: Strong: This paper deepens the understanding of some phenomenon or lowers the barriers to an existing research direction.

**Justification For Ethical Concerns:**

Since this is a work about text generation (text simplification) for the medical domain, it is important to discuss if the generated text could somehow cause any damages to the user, given that the users are inteded to be lay persons.

**Paper Topic And Main Contributions:**


The authors constructed the first multiligual simplication corpus for the medical domain. It includes four languages and consist of a clean (manually validated) and noisy (automatically compiled) datasets. Experiments with the datasets have been carried out using zero-shot and fine-tuning. A sample of the generated simplifed texts was manually chaked by exeprts. They provide a good discussion of the results and will make both the datasets and the human evaluation available for download.

**Questions For The Authors:**


1. Page 4, line 240: Please provide agreement also in terms of the Kappa coefficient.

2.- Page 4, lines 266-269: Please better explain this, that even though 60% of the sentences were removed, "most" of the information remained. How was this actually evaluated? Further, in lines 289-280, it is strange that the "simplified" sentences are actually larger than the original (complex) ones. Usually, longer sentences are more difficult to read than shorter ones. I understad that the simplified text can be longer, but I'd actually expect the sentences to be shorter.

3. I wonder why the authors did not try any LLM speciafically built for the medical (or biomedical) domain, such as SciFive. And I miss details about the hyperparameters that were used in the experiments. Will the source code be made available as well?

* minor comments:

- Page 3, line 16: I wonder if one of the difference to MUSS is that this was not built for the medical domain?

- Page 3, lines 225-231: 25+75=100, not 101 as stated earlier in this section.

- Page 4, line 321: Was "DP" previoulsy defined?

- Page 5, line 383: Did you try other values for the threshold?

- Page 6, lines 467-469: I think that this statement is only valid for the BLEU scores.




**Reasons To Accept:**

- The construction of a new resource that will be helpful for text simplification in the medical domain.
- Comprehensive experiments with the dataset, a good discussion of the results, and a short error analysis.
- Availability of the dataset and human evaluation.

**Reasons To Reject:**

- Lack of experiments with language models which were specifically built for the medical domain.
- Lack of details about the experiments, e.g., hyperparameters.

**Reproducibility:**

2: Would be hard pressed to reproduce the results. The contribution depends on data that are simply not available outside the author's institution or consortium; not enough details are provided.

**Reviewer Confidence:**

3: Pretty sure, but there's a chance I missed something. Although I have a good feel for this area in general, I did not carefully check the paper's details, e.g., the math, experimental design, or novelty.

---

> ### Author Rebuttal · Authors · 2023-08-28
>
> Thank you for your detailed review!
>
> **Re Question 1:** We have calculated the Krippendorff’s Alpha in regards to your question to be 0.662. We will add this to the paper.
>
> **Re Question 2:** We agree that we worded this particular sentence poorly.  We had meant to emphasize that the simple texts were a more condensed version of the complex texts. As for why simplified sentences may be longer than the complex sentence, this happens usually because technical sentences use jargon for brevity which has to be explained or elaborated upon in the simplified version. This usually involves succinctly explaining a certain disease, treatment, or research method to make the text more comprehensible to a lay reader.
>
> **Re Question 3 & LMs for medical domain:** At the time of our evaluation, we had not come across any medical-specific LLMs that were of comparable performance and had the same level of multilingual capabilities to the LLMs we chose to evaluate.
>
> **Re Question 3 & Hyperparameters:** We do provide hyperparameters for the experiments we performed under section 5.1 under Training Setup. Additionally, the source code will be provided in a github repo along with the dataset.
>
> **Re minor comments:** Thank you for these comments. The 75 was supposed to be a 76. We seemed to have overlooked this mistake, so thank you for catching it. We’ll make sure to fix these in the camera-ready version. As for your comment on page 5, we did assess several threshold values and observed a tradeoff between the noisiness and extractiveness of the resulting data, which is why we settled on the value of 0.5. As for your comment on page 6, we also observed an improvement in the BERTScore. However, we do concede that filtering did have a negative impact of the simplicity measure (SARI and LCS).
>
> **Re ethical questions:**
>
> We certainly agree that in this domain it necessary to be especially cognizant of the risks posed by  potential factual errors and hallucinations. Nonetheless, following a body of existing work on medical text simplification and/or summarization, we believe that this line of work is important both to advance technology for social good while also acknowledging the risks: (1) Inaccessibility of medical information causes problems in health literacy, which is one of the leading reasons for real health consequences including more hospital admissions, emergency room visits, and poorer overall health. Making medical information accessible is one of the best ways to tackle health literacy, and that is the core of what a simplification system is aimed to do. (2) Medical misinformation is one of the most series issues highlighted in the COVID-19 pandemic; one contributing reason for this is the lack of health literacy among the general public. By simplifying trustworthy evidence, we hope to empower the public with a keener eye for such misinformation. (3) Factual errors is one of the key aspects studied in this work; we perform a thorough evaluation dissecting issues that can come from these models, especially in a multilingual setting. We believe rigorous evaluation, as done in this work, is one of our best tools to demystify language models, and help the community understand the issues at hand. With such understanding, we hope to point to future directions that collectively, we will be able to provide factual, readable, multilingual access to medical texts.

---

### Official Review · Reviewer_gRpt · 2023-08-04

**Soundness:** 4

**Ethical Concerns:**

Yes

**Excitement:**

3: Ambivalent: It has merits (e.g., it reports state-of-the-art results, the idea is nice), but there are key weaknesses (e.g., it describes incremental work), and it can significantly benefit from another round of revision. However, I won't object to accepting it if my co-reviewers champion it.

**Justification For Ethical Concerns:**

The copyright status of the collected/distributed data from the Cochrane library is not discussed in the paper. The authors should simply indicate that they got Wiley’s authorization to build and distribute their corpus.

**Paper Topic And Main Contributions:**

This paper deals with text simplification in the medical domain, with the source text and the simplified text being in different languages. The authors propose a new corpus of paired sentences sourced from the Cochrane Library (English to another language among Spanish, French, Farsi).


**Questions For The Authors:**

Other important missing points

- the copyright status of the source material is not discussed. AFAIK, Wiley allows text mining, but redistributing a dataset built from Cochrane is another thing.

- The profile of the annotators involved in this work (alignement, evaluation) needs to be described (medical background, language fluency…). For the evaluation, a medical background is expected to check the factuality of the output. Yet, to evaluate the simplicity, one would expect someone with no medical background.


**Reasons To Accept:**

Positive points
The comparison of the automatic and human-based metrics is enlightening.
The proposed resource may be useful for further research.
The proposed approach to build the corpus, combining existing techniques and leading to a gold corpus and a silver one, is sound.


**Reasons To Reject:**

Negative points

- the whole problem is ill-justified. The simple approach consisting in simplifying and then translating (one model to simplify from En to En, and a translation model from En to the target language) is rejected in a footnote, stating that some languages are not correctly dealt with by translation systems. Yet, the whole paper only considers very well covered languages (with the exception of Farsi maybe, although it is not a low resource language), and systems (GPT, T5…) that were trained on mostly English and some major languages.

- Following the previous point: no comparison is made with such simplify-and-then-translate systems. The results of GPT3_zero on En-to-En tend to indicate that translating the output would certainly be a good solution.

- As noted by the authors, taking the sentence as the atomic unit is not satisfying; simplifying may imply splitting or grouping sentences. What is the reason of this choice? Modern models, such as LLM, can process texts with more context than sentence by sentence.

**Reproducibility:**

4: Could mostly reproduce the results, but there may be some variation because of sample variance or minor variations in their interpretation of the protocol or method.

**Reviewer Confidence:**

4: Quite sure. I tried to check the important points carefully. It's unlikely, though conceivable, that I missed something that should affect my ratings.

**Typos Grammar Style And Presentation Improvements:**


r is presented as a function (ratio of overlapping tokens…) and a threshold with a fixed value. Give a different name to the 2 objects.

---

> ### Author Rebuttal · Authors · 2023-08-28
>
> Thank you for your feedback!
>
> **Re simplify-then-translate systems:** We do evaluate the simplify-then-translate pipeline using GPT-3 and Google Translate and it does tend to outperform the multilingual pipeline in our evaluation. Our methodology is described in section 5.1 under zero-shot models, and the results can be found in Table 4 and Table 5 with the system name GPT_trans. However, the motivation for preferring the multilingual pipeline over the simplify-then-translate pipeline stands: Simplify-then-translate requires two models, and is more expensive (when scaled up) in terms of cost and compute compared to the multilingual pipeline. The multilingual pipeline also addresses some of the robustness issues associated with machine translation as described in the papers we cited. Ruder et al. (2021) described the poor performance of MT models in low-resource languages, but it also describes poor performance in languages with non-Latin scripts. Vu et al. (2022) discussed how using machine translation along with tasks like cross-lingual summarization leaves significant headroom for improvement. Based on this, we believe the multilingual pipeline has a greater potential for improvement compared to the simplify-then-translate approach. This paper is a step in the direction of solving what we see as an open problem.
>
> **Re atomic unit sentences:** The main reason for this choice was an effort to try to constrain the models to generate sentence-to-sentence simplifications. Accounting for the splitting and the grouping could have caused models trained on these alignments to generate variable length simplifications, which would have been hard to evaluate against gold labels. This combined with the possible complications from using multilingual data, made the sentence-level approach a reasonable starting point from our view. However, as we explain in the limitations section, we do think that working on larger units makes sense.
>
> **Re Copyright status:**  the authors have had a fair use discussion with our institution library and copyright office about this: the research use of publicly available abstracts, including both technical abstracts and multi-lingual plain language summaries, is considered fair use. This work does not use any full texts of Cochrane reviews. Note that the Cochrane dataset, which contains only abstracts, was released publicly by prior work (Devaraj et al 2021 “Paragraph Level Simplification of Medical Texts”, Guo et al 2021 “Automated Lay Language Summarization of Biomedical Scientific Reviews”). Our institution library recommends the Creative Commons license for this work. We will clarify this in the paper as well as the repo for data release.
>
> **Re annotator profile:** The annotators were all fluent in the language they were assigned to annotate. They did not have medical backgrounds, but they did have a strong background in research and in evaluating technical texts. Since factuality was assessed by determining how factually consistent the simplification was to the original input, a medical background was not a necessity. If the annotators could comprehend both the input and the simplification, expert knowledge is not really required to make such an assessment. Annotators were also always encouraged to factcheck anything they were unsure about. We will add this information to the paper in future drafts.

---

### Official Review · Reviewer_XHER · 2023-08-04

**Soundness:** 4

**Ethical Concerns:**

Yes

**Excitement:**

4: Strong: This paper deepens the understanding of some phenomenon or lowers the barriers to an existing research direction.

**Justification For Ethical Concerns:**

Following the Ethics Review questions for papers presenting new datasets, the authors should address these questions in their paper:

1. Describe how intellectual property (copyright, etc) was respected in the data collection process.

2 Describe how crowd workers or other annotators were fairly compensated and how the compensation was determined to be fair.

**Missing References:**

The literature review is correct and up to date.


**Paper Topic And Main Contributions:**

The paper falls under the "new data resources" topic.

The main hypothesis emphasises the importance of simplifying complex medical texts to make them understandable for the general public, facilitating access to critical medical information.

Text simplification models have emerged to address this issue, automatically transforming complex texts into simpler versions that readers can understand without specific healthcare training. However, these models have focused solely on simplifying texts in a single language. This limits the availability of simplified information in multiple languages, especially in the medical domain, leading to inequalities for those who do not speak English and face barriers to accessing information.

The paper's primary contribution is presenting MULTICOCHRANE, the first parallel dataset for medical text simplification across multiple languages: English, Spanish, French, and Farsi. MULTICOCHRANE contains aligned sentence pairs sourced from the Cochrane Library of Systematic Reviews, a repository of meta-analyses of treatment effectiveness. These review articles consist of both technical abstracts and plain-language summaries (PLS), from which we derive two subsets of MULTICOCHRANE:

* MC-CLEAN contains 101 technical abstracts with manually aligned sentence pairs by experts.
* MC-NOISY, a larger but noisier subset created using an automatic sentence alignment model.

The authors evaluate various pre-trained language models on text simplification in four languages and conduct human evaluations regarding simplicity, fluency, and factuality. The results demonstrate that while pre-trained models effectively simplify English, their performance significantly degrades in other languages, leading to factuality errors. Additionally, the approach of translating English simplifications to other languages is analysed, which, in many cases, can circumvent these issues.

Overall, this resource is novel, multilingual, and thoroughly evaluated, and it can help stimulate further efforts to improve the performance of models across different languages.


**Questions For The Authors:**

A. What predictions do you have about the results for non-Indo-European languages, such as Chinese or Japanese, where sentence alignment is more complex?

B. It is mentioned that sentence-level simplification has some limitations, especially in the medical field, where contextual information is important. How could the paragraph or discourse be incorporated into the alignment methodology?

C. How could insights from human evaluations be incorporated into the evaluation metrics of the models?


**Reasons To Accept:**

The most important contributions of the paper:

1. Advancing Multilingual Medical Text Simplification: The paper addresses the limitation of existing work, focusing solely on monolingual text simplification. It presents MULTICOCHRANE, the first parallel dataset for medical text simplification across 4 Indoeuropean languages, English, Spanish, French, and Farsi, with a range of speakers from 80 million to more than 400 million. This advancement enables the direct simplification of complex texts into target languages.

2. Dataset Creation: The authors introduce the MC-CLEAN subset, consisting of 101 technical abstracts with expert-annotated manual alignments in English, then semi-automatically aligned to other languages, totalling around 5,000 sentence pairs across all four languages. They also present the MC-NOISY subset, a larger but noisier dataset created using an automatic sentence alignment model trained on MC-CLEAN, containing approximately 100,000 sentence pairs across languages.

3. Systematic Evaluation of Simplification Models: The authors evaluate various large pre-trained language models in zero-shot and fine-tuned settings on text simplification across four languages.

4. Human Assessments: The study includes human assessments covering simplicity, fluency, and factuality, in addition to automatic evaluations. The correlation between automatic metrics and human assessments is analysed, revealing mostly weak correlations.

5. Identifying Model Performance Differences: The results show that while pre-trained models effectively simplify English, their performance significantly degrades in other languages, leading to factuality errors. Different models, such as GPT-3 (zero-shot) and Flan-T5 (zero-shot), exhibit varying strengths and weaknesses when simplifying to different languages.

6. Availability of Resources: The authors publicly release MULTICOCHRANE, model outputs, and all human judgments collected during the study, motivating future research on multilingual medical text simplification to enhance model performance across those four languages.

In conclusion, this study contributes to multilingual medical text simplification by introducing a novel and evaluated dataset, providing insights into model performance across languages, and releasing valuable resources for further research and development in this area. However, it also acknowledges certain limitations, such as uneven distribution of multilingual versions for articles and potential drawbacks of sentence-level simplification in the medical domain. The discussion of the results of the evaluation of the models is very relevant and contributes to the final quality of the article.

The relevance of the appendices should also be mentioned: they convincingly demonstrate the quality of the work.


**Reasons To Reject:**

There are no strong reasons against the paper, only suggestions for improving the resource and the methodology.


* Limitation of Languages: Although MULTICOCHRANE covers several languages (English, Spanish, French, and Farsi), it is acknowledged that there is an uneven distribution of resources, and some languages, such as Farsi, may have less coverage of multilingual articles. This could bias the results towards languages with more resources and affect the representativeness. It would also be interesting to extend the approach to other non-Indo-European languages, such as Chinese and Japanese, to compare results.

* Weakly Correlated Human Evaluation: It is mentioned that the correlation between automatic metrics and human evaluations is mainly weak. This could suggest that the automatic metrics used to evaluate the models do not fully capture the quality of the simplifications, raising questions about the validity and robustness of the conclusions. Let us not forget that human evaluations of the quality of the results are more important.

* Quality Degradation in Other Languages: The results show that pre-trained models perform significantly less when simplifying texts in languages other than English, introducing accuracy errors. This quality degradation could affect the models' practical utility in multilingual applications.

Overall, these problems may suggest that the study has certain limitations, especially in practical application. Still, the quality of the analysis of the evaluations and the methodology employed is beyond doubt.


**Reproducibility:**

5: Could easily reproduce the results.

**Reviewer Confidence:**

3: Pretty sure, but there's a chance I missed something. Although I have a good feel for this area in general, I did not carefully check the paper's details, e.g., the math, experimental design, or novelty.

**Typos Grammar Style And Presentation Improvements:**

Overall the paper is well-written and well-structured.

---

> ### Author Rebuttal · Authors · 2023-08-28
>
> Thank you for your feedback!
>
> **Re languages:** yes, exploring the effect of the imbalanced distribution and including more languages are excellent directions for future work!
>
> **Re automatic evaluation metrics:** we absolutely agree! One takeaway from this paper is a call for the development of more robust automatic metrics.
>
> **Re quality degradation in other languages:** we believe that this points to the weakness in LLMs that claim to be multilingual, hence revealing a research gap in the training and evaluation of LLMs.
>
> **Re A:** The amount of data collected for the other languages was significantly lower compared to the amount we collected for the languages we used for evaluation. This, plus potentially noisier alignments with those languages, will likely lead to worse results. However, the other factor is also whether that language is high-resource in the training data of the underlying multilingual LLMs, which is a separate distribution than the amount of data available in Cochrane (e.g., Cochrane has more Farsi data, but that may not be true when the underlying LLM was trained in the general domain). As you mentioned in your review, extending our evaluation to non-Indo-European languages is definitely a worthy future direction. As a first step in this direction however, we decided to focus our evaluation to the languages that had the most data available due to resource and time constraints.
>
> **Re B:** While the CRF method we used for alignment is contextual, we had null-alignments when sentences were deleted from the original text or added to the simplified text. Thus, future efforts for alignments could consider the role of such deletions and elaborations in the discourse, and incorporate those into the alignment itself.
>
> **Re C:** The human evaluation results show that automatic metrics like BLEU and SARI lack the level of understanding required to reliably evaluate simplifications. Metrics for evaluating simplifications need to be able to better assess factuality and simplicity.
>
>
> **Re intellectual property:** the authors have had a fair use discussion with our institution library and copyright office about this: the research use of publicly available abstracts, including both technical abstracts and multi-lingual plain language summaries, is considered fair use. This work does not use any full texts of Cochrane reviews. Note that the Cochrane dataset, in similar capacity as ours, has already been released publicly by prior work (Devaraj et al 2021 “Paragraph Level Simplification of Medical Texts”, Guo et al 2021 “Automated Lay Language Summarization of Biomedical Scientific Reviews”). Our institution library recommends the Creative Commons license for this work. We will clarify this in the paper as well as the repo for data release.
>
> **Re annotator compensation:** All annotators are hired as hourly research assistants by our institution, with a pay of $15/hr. We will add this information to the paper.

---

### Meta-Review · Area_Chair_SAf6 · 2023-09-19

**Recommendation:** 4

**Metareview:**

Review Summary:
Reviewers agree that the paper advances multilingual medical text simplification, creates a new dataset, and evaluates various pre-trained language models in zero-shot and fine-tuned settings are positive contributions. Some reasons to reject this paper are stated including not using language models specifically trained for the medical-domain, poor justification of approach, poor human evaluation quality, degradation of performance on certain languages, and lack of detailed information regarding experiments. In addition, since this paper presents a generative approach for the medical domain, it raises additional ethics consideration including the intellectual property regulations, annotators compensation, etc

Reasons to Accept:
(1) The paper makes significant contributions to multilingual medical text simplification, introducing the MULTICOCHRANE dataset for four Indo-European languages.
(2) It presents a novel approach to creating the dataset, including MC-CLEAN and MC-NOISY subsets, which can be valuable resources for further research.
(3) The systematic evaluation of simplification models in both zero-shot and fine-tuned settings across multiple languages provides insights into model performance and challenges.
(4) The inclusion of human assessments alongside automatic evaluations, and the analysis of their correlation, enhances the understanding of model performance.
(5) The paper identifies differences in model performance when simplifying to different languages, shedding light on potential quality degradation.
(6) The release of MULTICOCHRANE, model outputs, and human judgments promotes future research in multilingual medical text simplification.

Reasons to Reject:
(1) While there are no strong reasons to reject the paper, some suggestions for improvement include addressing the limitation of uneven language distribution in the dataset, exploring the use of the simplify-and-translate approach for comparison, and considering more context-aware approaches instead of sentence-level simplification. Additionally, providing more experimental details and experiments with domain-specific language models could enhance the paper's quality.

The author rebuttal have tried to address these issues.

---

### Decision · Program_Chairs · 2023-10-07

**Decision:**

Accept-Main

**Comment:**

Review Summary:
Reviewers agree that the paper advances multilingual medical text simplification, creates a new dataset, and evaluates various pre-trained language models in zero-shot and fine-tuned settings are positive contributions. Some reasons to reject this paper are stated including not using language models specifically trained for the medical-domain, poor justification of approach, poor human evaluation quality, degradation of performance on certain languages, and lack of detailed information regarding experiments. In addition, since this paper presents a generative approach for the medical domain, it raises additional ethics consideration including the intellectual property regulations, annotators compensation, etc

Reasons to Accept:
(1) The paper makes significant contributions to multilingual medical text simplification, introducing the MULTICOCHRANE dataset for four Indo-European languages.
(2) It presents a novel approach to creating the dataset, including MC-CLEAN and MC-NOISY subsets, which can be valuable resources for further research.
(3) The systematic evaluation of simplification models in both zero-shot and fine-tuned settings across multiple languages provides insights into model performance and challenges.
(4) The inclusion of human assessments alongside automatic evaluations, and the analysis of their correlation, enhances the understanding of model performance.
(5) The paper identifies differences in model performance when simplifying to different languages, shedding light on potential quality degradation.
(6) The release of MULTICOCHRANE, model outputs, and human judgments promotes future research in multilingual medical text simplification.

Reasons to Reject:
(1) While there are no strong reasons to reject the paper, some suggestions for improvement include addressing the limitation of uneven language distribution in the dataset, exploring the use of the simplify-and-translate approach for comparison, and considering more context-aware approaches instead of sentence-level simplification. Additionally, providing more experimental details and experiments with domain-specific language models could enhance the paper's quality.

The author rebuttal have tried to address these issues.